# Benchmarking of analysis strategies for data-independent acquisition proteomics using a large-scale dataset comprising inter-patient heterogeneity

Klemens Fröhlich [1,2,3,13], Eva Brombacher [2,3,4,5,13], Matthias Fahrner [1,2,3], Daniel Vogele [1,2], Lucas Kook[6,7], Niko Pinter[1], Peter Bronsert [1,8,9], Sylvia Timme-Bronsert[1,9], Alexander Schmidt [10], Katja Bärenfaller [11], Clemens Kreutz [4,5,14] & Oliver Schilling [1,8,12,14✉]

Numerous software tools exist for data-independent acquisition (DIA) analysis of clinical samples, necessitating their comprehensive benchmarking. We present a benchmark dataset comprising real-world inter-patient heterogeneity, which we use for in-depth benchmarking of DIA data analysis workflows for clinical settings. Combining spectral libraries, DIA software, sparsity reduction, normalization, and statistical tests results in 1428 distinct data analysis workflows, which we evaluate based on their ability to correctly identify differentially abundant proteins. From our dataset, we derive bootstrap datasets of varying sample sizes and use the whole range of bootstrap datasets to robustly evaluate each workflow. We find that all DIA software suites benefit from using a gas-phase fractionated spectral library, irrespective of the library refinement used. Gas-phase fractionation-based libraries perform best against two out of three reference protein lists. Among all investigated statistical tests non-parametric permutation-based statistical tests consistently perform best.

[1] Institute for Surgical Pathology, Medical Center—University of Freiburg, Faculty of Medicine, University of Freiburg, Freiburg im Breisgau, Germany. [2] Faculty of Biology, University of Freiburg, Freiburg im Breisgau, Germany. [3] Spemann Graduate School of Biology and Medicine (SGBM), University of Freiburg, Freiburg im Breisgau, Germany. [4] Institute of Medical Biometry and Statistics, Faculty of Medicine and Medical Center – University of Freiburg, Freiburg im Breisgau, Germany. [5] Centre for Integrative Biological Signaling Studies (CIBSS), University of Freiburg, Freiburg im Breisgau, Germany. [6] Epidemiology, Biostatistics & Prevention Institute, University of Zurich, Zurich, Switzerland. [7] Institute for Data Analysis and Process Design, Zurich University of Applied Sciences, Winterthur, Switzerland. [8] German Cancer Consortium (DKTK) and German Cancer Research Center (DKFZ), Heidelberg, Germany. [9] Tumorbank Comprehensive Cancer Center Freiburg, Medical Center University of Freiburg, Freiburg im Breisgau, Germany. [10] Proteomics Core Facility, Biozentrum, University of Basel, Basel, Switzerland. [11] Swiss Institute of Allergy and Asthma Research (SIAF), University of Zurich, and Swiss Institute of Bioinformatics (SIB), Wolfgang, Switzerland. [12] BIOSS Centre for Biological Signaling Studies, University of Freiburg, Freiburg im Breisgau, Germany. [13] These authors contributed equally: Klemens Fröhlich, Eva Brombacher. [14] These authors jointly supervised this work: Clemens Kreutz, Oliver Schilling. ✉email: oliver.schilling@uniklinik-freiburg.de

Proteomics denotes the study of the entire set of proteins produced by an organism under defined conditions. While the genome of an organism is geared towards remaining static for almost every cell, the dynamics introduced by the proteome, including differential expression, altered activity, and modifications of proteins, allow cells, tissues and even the whole organism to undergo dramatic changes and to carry out a plethora of different functions. Often, the term 'proteomics' is specifically used to refer to large-scale studies of the proteome employing liquid chromatography (LC) coupled to tandem mass spectrometry (LC-MS/MS).

Many studies, e.g., in the clinical context, focus on the detection of differentially abundant proteins, preferably on a proteome-wide scale[1,2]. To identify such proteins, modern mass spectrometry-based proteomics techniques offer many ways to quantify and compare proteins between samples[3]. Due to their simplicity and cost-effectiveness, label-free approaches have been used for decades. Historically, label-free samples were measured using data-dependent acquisition (DDA). In DDA, following a survey scan, masses of interest are selected for further fragmentation based on their intensity. This allows for narrow isolation windows and results in fragment spectra of low complexity. However, the fact that the masses of interest are selected during the measurement introduces stochastic sampling effects[4].

In contrast, parallel fragmentation of precursor ions implemented in data-independent acquisition (DIA) methods is independent of ion intensity and other properties, leading to constant data acquisition between samples[5]. In DIA proteomics, quantification is typically performed on the fragment level and not at the precursor level as in the case of DDA. The assignment of fragment ions to a single analyte (i.e., peptide) happens post-measurement and is strongly dependent on properties of reference spectral libraries as well as on features of the data processing algorithms[6–8].

Spectral libraries for DIA contain peptide reference data, i.e., mass-to-charge ratio, retention time and/or fragment spectra. Typically, only those peptides which are present in the spectral library can be detected in DIA analyses, highlighting the importance of spectral library generation. The National Institute for Standards and Technology (NIST) is developing generally applicable reference mass spectral libraries. Yet, project-specific spectral libraries tend to be in more widespread use since they may better reflect individual properties of a certain mass spectrometer as well as the overall proteome composition of the specific samples under investigation. Spectral libraries predicted in silico are currently gaining momentum[9]. Project-specific libraries can be obtained from analyzing DDA runs of samples, which are representative of the project, or by refining an in silico predicted spectral library. Spectral libraries can be refined e.g., by gas-phase fractionation (GPF), where a single sample is repeatedly measured to investigate distinct mass-to-charge ranges in greater detail.

While its complexity makes the handling and analysis of DIA data more laborious, it has been demonstrated that the quantification by DIA is more robust compared to DDA[10,11]. Recently, DIA has reached a protein coverage that is comparable to, or even exceeds, the one of DDA[12,13]. It has also been shown that for the analysis of post-translational modifications DIA can outperform DDA, for example for phosphoproteomics analyses[14–16].

To objectively compare data processing and quantitation methods, the proteomics community often employs so-called 'spike-in benchmark datasets', where peptides with known properties (e.g., sequence and concentration) are added to 'background' peptides with likewise known properties. To mimic the complexity encountered in realistic settings often different organisms are combined to create benchmark datasets. These benchmark datasets are valuable tools for controlling and optimizing different aspects of data acquisition and analysis, including LC-MS/MS parameters[8,17], library generation[17,18], analysis software parameters[19], data preprocessing, and statistical analysis[20] for detecting differentially abundant proteins. The critical importance of data processing in DIA proteomics renders benchmarking datasets particularly useful for this methodology.

Benchmark studies published to date have mainly focused on technical reproducibility and data acquisition[7,8], or on individual data analysis steps, such as data preprocessing in the form of data normalization or data imputation, and statistical methods[20–22]. Indeed, the downstream analysis of the data acquired by DIA software suites should be carefully reflected upon, going beyond peptide-spectrum-matching (PSM) and quantitative signal/feature integration. Furthermore, valid benchmarking datasets should represent inter-individual heterogeneity on a scale that is comparable to present-day, cohort-wide proteome studies, which is rarely the case.

Hence, especially in biomarker discovery studies in which highly heterogeneous patient proteomes are investigated, current benchmark datasets provide little help for a user who has to decide whether and how to generate a library for DIA analysis, which tool to use for the DIA analysis itself, and, most importantly, how this will affect data preprocessing and statistical analysis for differentially abundant proteins.

Here, using a benchmark dataset reflecting real inter-individual heterogeneity we show that the choice of library generation and DIA software affect data properties, such as data sparsity, and how data preprocessing and statistical analysis methods affect the identification of differentially expressed proteins. We assess this by means of objective evaluation measures based on p-values and log2 fold-changes that result from each investigated analysis workflow.

## Results and discussion

**Significance of benchmark studies for the field of proteomics.** Benchmark studies have become an invaluable tool to objectively assess the advantages and disadvantages of the choices made over the course of proteomics studies, including the choice of sample preparation, data acquisition, MS data analysis and statistical processing. However, benchmark studies often suffer from small sample sizes and unrealistically low background variance[7,23,24].

Yet, biomarker discovery studies often include hundreds of patients with heterogeneous proteomes and can contain high within- and between-person variance. Results acquired on common benchmark datasets not considering such heterogeneity may not be ideal to guide a researcher to the best suited analysis workflow in a clinical setting.

Therefore, we set out to create a large benchmark dataset reflecting real inter-individual heterogeneity to investigate the interplay between library generation, DIA software analysis, data preprocessing, and statistical analysis. To this end, we acquired sentinel lymph nodes from multiple patients as formalin-fixed paraffin-embedded tissue (FFPE) and used *Escherichia coli* (*E. coli*) peptides as a spike-in peptide subpopulation of known concentration.

The experimental design of our study aims at investigating highly complex human samples, i.e., tissues. Other proteomes such as plasma proteomes may display a lower complexity and may require dedicated benchmarking[25].

Following protein digestion and sample clean-up, we split the samples into four groups and added *E. coli* peptides in *E. coli* to human peptide ratios of 1:6, 1:12, and 1:25, or did not add *E. coli* peptides at all ('human only'). The spike-in concentrations were chosen in such a way that the measure of performance can be

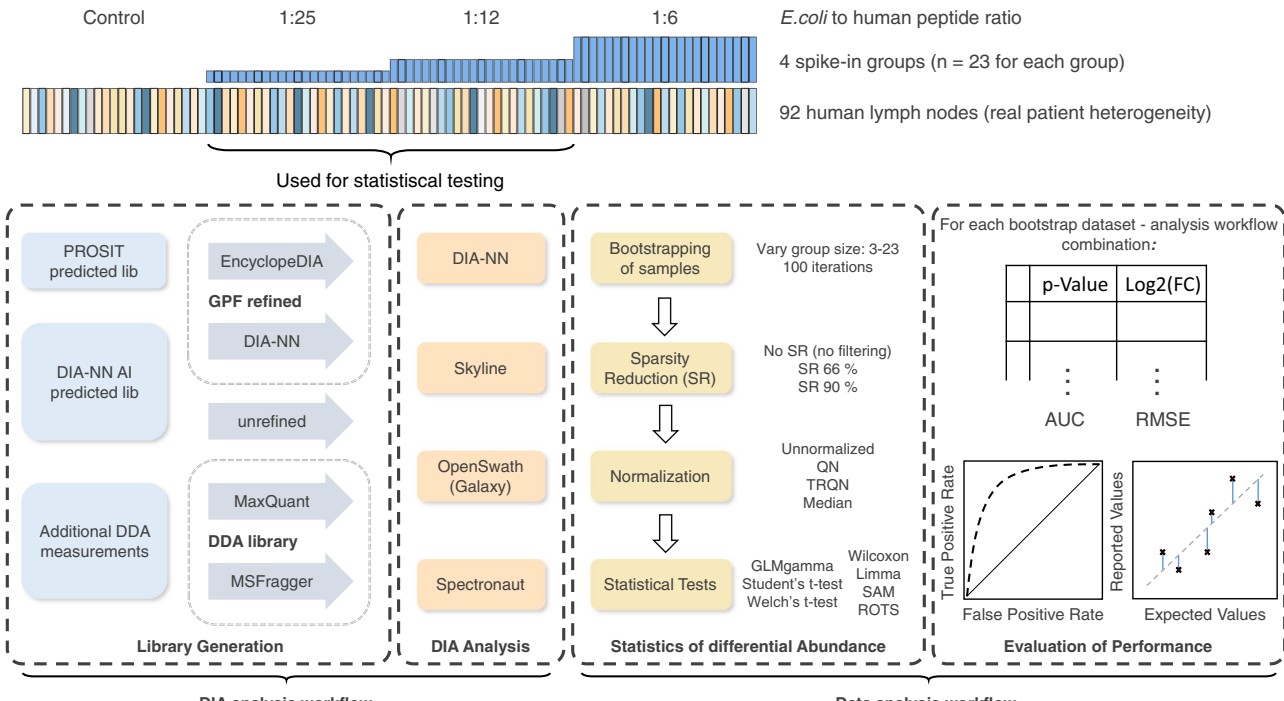

**Fig. 1 Benchmarking workflow.** A data-independent acquisition (DIA) benchmark dataset was created by adding *E. coli* peptides in known ratios to peptide preparations of lymph nodes of 92 individuals. We analyzed the raw data with different spectral libraries and DIA software suites. From samples to which *E. coli* peptides were added in the two *E. coli*: human peptide ratios 1:25 and 1:12, bootstrap datasets with group sizes of 3 to 23 were generated. For each of those 21 different group sizes, 100 bootstrap datasets were generated. On each bootstrap dataset different data analysis workflows, composed of different sparsity reductions, normalization options, and different statistical tests for detecting differentially abundant proteins, were applied. The results were returned in a table containing *p*-values and log2 fold-changes (log2FCs) for each protein. As the ground truth about the changed proteins (*E. coli*) is known, the prediction performance of each workflow can be assessed. This can be done based on the *p*-values from the statistical tests by calculating the receiver operating characteristic (ROC) curve, based on which the area under curve (AUC) is calculated. To quantify the accuracy of quantification the root-mean-square error (RMSE) is calculated based on the detected log2FC.

based on a sufficient number of identified *E. coli* proteins (aiming for at least 100 identified proteins even in the smallest spike-in condition). Those four groups are referred to as 'spike-in conditions' and have a size of $n = 23$ each (Fig. 1).

The *E. coli* proteome itself comprises a wide dynamic range of protein concentrations and we expect different DIA software-library combinations to differ in their ability to detect and quantify low abundant *E. coli* proteins. We can objectively assess those differences as the relative concentration of *E. coli* proteins is known for all spike-in conditions.

In comparison to defined protein mixtures such as the Universal Proteomics standard (UPS1, Sigma), the *E. coli* proteome may be regarded as a more trustworthy proxy of the natural proteome complexity. We believe that our study design is a good trade-off between a sufficiently complex proteomic ground truth and the preservation of the intrinsic heterogeneity of human samples.

As DIA has been shown to outperform DDA in different settings, we chose to employ DIA for the measurement of our samples, using an established acquisition scheme[26]. The resulting set of DIA LC-MS/MS measurements (92 LC-MS/MS files) consists of 12.4 million MS2 spectra.

**Generation of spectral libraries.** Three trends can be observed in current DIA analysis strategies: (a) using spectral libraries generated by analyzing pre-fractionated DDA runs[27], (b) using spectral libraries generated by refining predicted libraries using GPF[17], or (c) using no additional experimental data to generate spectral libraries (e.g., using predicted libraries)[9,28].

All of these prototypical approaches are integrated in this study. We chose to measure our pre-fractionated samples using DDA, as this allows us to integrate well-established DDA analysis tools such as MaxQuant and MSFragger. However, it would also be possible to measure the pre-fractionated master mix using a DIA method to refine a predicted library in this manner.

Combining spectral library analysis approaches with DIA data analysis software led to 17 different 'DIA workflows'. We used standard parameters, opting for recommended settings whenever possible, to reflect a realistic average user scenario and to prevent over-optimization. The resulting 17 DIA workflow datasets were then combined with 'data analysis workflows' combining bootstrapping with three sparsity reduction methods, four normalization methods, and seven statistical test options resulting in 1428 analysis combinations.

To generate experiment-specific spectral libraries, we performed GPF on a mastermix, which represents an average spike-in concentration of *E. coli* to human peptides of 1:15. Using DIA-NN to refine an in silico predicted DIA-NN spectral library of *E. coli* and human proteins, we generated a spectral library containing 84016 precursor entries mapping to 10459 proteins. Using an in silico predicted PROSIT spectral library refined by EncyclopeDIA, we generated a spectral library containing 45445 precursors mapping to 8472 proteins[18].

We also pre-fractionated a master mix to obtain samples for in-depth DDA library generation[29]. By applying Fragpipe to the resulting DDA files we generated a spectral library containing 81409 precursors mapping to 7781 proteins. We also used MaxQuant to build a DDA-based spectral library containing 51260 precursors mapping to 7382 proteins.

For the spectral library generation, iRT peptides, a well-established retention time standard, were used to guide the retention time alignment[30]. It is, however, also possible to perform retention time alignment without adding additional peptides (e.g., DIA-NN does not rely on iRT peptides).

Using DIA-NN in combination with the in silico predicted DIA-NN GPF-refined spectral library, on average 48,698 precursors were identified per measurement with an average chromatographic peak width of 8 s (full width at half height). To avoid batch effects originating from sample preparation or order of measurement we randomly assigned the lymph nodes to each spike-in condition. Additionally, we used block randomization during the data acquisition[31]. To avoid carry-over of *E. coli* peptides, we measured the conditions in order of ascending *E. coli* concentration. This restriction of random measurement was necessary to keep *E. coli* peptides from appearing in the 'control' spike-in condition. No batch effects stemming from sample preparation or measurement order could be detected in this study (Supplementary Fig. 1).

**Study design for the assessment of data analysis workflows**. For our study we selected four commonly used DIA software analysis suites: DIA-NN[32], Skyline[33], OpenSwath[34], and Spectronaut[10]. Whenever possible, we combined all generated libraries with all DIA analysis software solutions (especially predicted spectral libraries in combination with the high number of samples are challenging for some software suites). We also included 'Direct-DIA', a feature of Spectronaut which does not require any additional experimental evidence for library generation. This also holds true when directly using a predicted library without refinement (here: DIA-NN predicted library used in combination with DIA-NN). This resulted in a total of 17 different DIA analysis workflows. It should be noted that the way peptide information is summarized to protein information differs between the investigated software suites. Thus, the number of proteins identified by different software suites cannot be directly compared. For all subsequent analysis steps, protein-level output from the DIA analysis workflows was used. However, some analyses have been also conducted at the precursor level and can be found in the Supplementary information file. For the sake of simplicity and relevance we focused our statistical analyses on the comparison between the two lowest *E. coli* spike-in conditions, 1:25 and 1:12, at the protein level (Fig. 1). This comparison poses the greatest challenge to any DIA analysis software as quantitations are usually less precise for low abundant proteins[27,35].

To get a robust estimate on the overall ability of the data analysis workflow to detect differentially abundant proteins as well as to investigate the effect of sample size, normalization, sparsity reduction, and choice of a statistical test using bootstrapping, we drew various subsamples of different sizes from each of the 17 original benchmark datasets. The analysis workflows are applied to the resulting bootstrap datasets, which show varying data characteristics, e.g., differing missing value percentages or median protein variances. By evaluating the different workflows on these bootstrap datasets our results become more robust and generalizable.

In brief, we randomly drew samples from each of the two lowest *E. coli* spike-in conditions with group sizes of three to 23 samples. To each bootstrap dataset, we applied different data analysis workflows composed of multiple options for the preprocessing steps in the form of sparsity reduction and normalization, followed by one of seven statistical tests to identify differentially abundant proteins.

Taking into account the aforementioned 17 different types of LC-MS/MS data processing, we acquired prediction performance information for 1428 different analysis workflows, each of which was applied to 2100 bootstrap datasets resulting in almost 3 million analyses.

This staggering number illustrates the amount of possible combinations of library generation methods, DIA software suites, and downstream data preprocessing and statistical analysis methods proteomics scientists are confronted with. As every study is different and there are no truly universally applicable methods available in proteomics, the level of experience and choices of the proteomics data analyst determine the reliability and reproducibility of a proteomics study, which was impressively demonstrated by Choi et al.[24].

**Descriptive analysis of LC-MS/MS benchmark dataset**. For each DIA analysis workflow we first assessed the number of identified and quantified proteins after a 1% protein FDR cutoff had been applied (Fig. 2, left, see Supplementary Figs. 10 and 11 for the overlap between the proteins that have been detected by the 17 DIA workflows). The DDA spectral libraries derived from high pH reversed-phase fractionated samples consistently led to smaller numbers of identified proteins. As the tissue used in this study had been formalin-fixed, chemical modifications can reduce the number of identified peptides and proteins during spectral library generation. In our experience, GPF-refinement of spectral libraries often increases the identification rates for DIA-type proteomics data of FFPE tissue.

In general, the total number of identifiable *E. coli* proteins increases with increasing spike-in concentrations, i.e., with increasing amounts of physically present *E. coli* proteins. Using a GPF-refined in silico predicted DIA-NN spectral library in combination with DIA-NN resulted in the highest number of quantified proteins.

However, in quantitative proteomics, protein identifications only serve a useful purpose if they are accompanied with robust and reliable quantitation. When summarizing the protein abundances as calculated by the different DIA analysis workflows both the shape of the distribution of log2-transformed protein abundances (Fig. 2, center) as well as the correlation of log2-transformed intensities between DIA analysis workflows (Supplementary Fig. 2) mostly depend on the choice of DIA analysis software, and to a lesser extent on the spectral library.

Further, we determined the variance of *E. coli* protein intensities per spike-in ratio and found it to be similar across all DIA analysis workflows (Fig. 2, right). Since one single batch of *E. coli* was used for all spike-in conditions, which reduces the inter-sample variability of *E. coli* proteins, a small variance indicates a reproducible quantitation algorithm. The number, distribution, and variance of identified precursors can be found in Supplementary Fig. 3.

We, furthermore, investigated the average number of reported quantitations for *E. coli* and human proteins per spike-in condition (Supplementary Fig. 4 and 5, respectively). For all DIA analysis software suites, the number of identified and quantified *E. coli* proteins decrease in lower spike-in conditions. We noticed that OpenSwath requires the usage of the TRIC tool for cohort-wide retention time alignment and chromatographic peptide filtering to yield the expected decrease of identified and quantified *E. coli* proteins in lower spike-in conditions (Supplementary Fig. 7a)[36]. To this end, we have now integrated the TRIC tool into the Galaxy environment. TRIC uses a graph-based alignment strategy on peptide fragment level to integrate information from all available runs, which leads to a reduced identification error. If using PyProphet FDR filtering in Open-Swath without TRIC, the numbers of identified and quantified *E. coli* proteins are highly similar across all spike-in ratios.

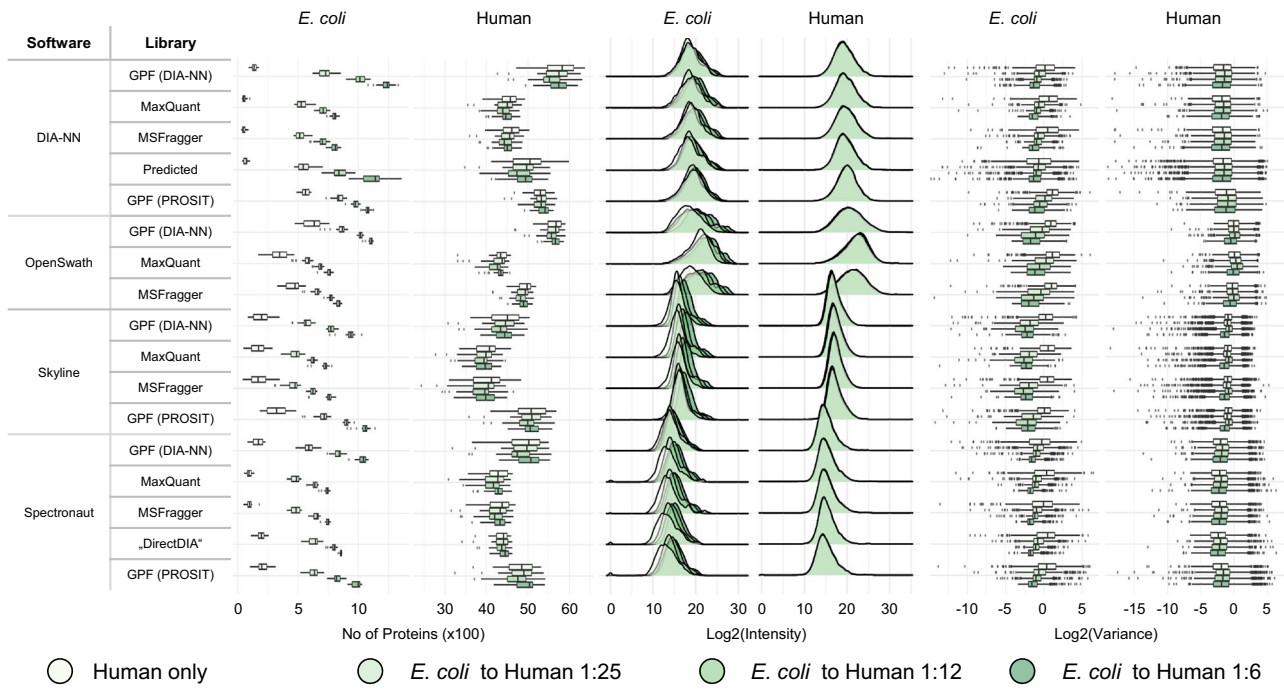

**Fig. 2 Choice of spectral library and DIA analysis software influences number of identified proteins.** Protein number, distribution, and variance for each DIA analysis workflow separated by species and color-coded by spike-in condition. Left: Number of all identified and quantified proteins in all 92 samples. Center: Log2 intensity distributions. Right: Log2 variance. Log2 variance values < −12 were excluded from this plot. For spike-condition, 1:6 data of $n = 22$ biologically independent samples have been used and for each of the other spike-in conditions data of $n = 23$ biologically independent samples have been used. The boxplots show median (center line), interquartile range (IQR, extending from the first to the third quartile) (box), and 1.5 * IQR (whiskers).

DIA-type proteomics promises to reduce the number of missing values, i.e., the missingness, in multi-sample proteomic experiments[10]. In the present dataset, 25% of all samples are human-only and void of *E. coli* proteins. This experimental setting not only supports the illustration of missingness but also the illustration of false-positive quantitation of proteins (here: *E. coli* proteins being found in human-only spike-in condition).

As can be appreciated from Fig. 3a, for all DIA software suites, there is a slight negative correlation between the means of the human and *E. coli* protein intensities and the percentage of missing values per protein, i.e., proteins with larger average intensities have less missing values (see Supplementary Fig. 8 for separate plots for each species). This negative correlation has been reported previously in a clinical proteomics study employing a tripleTOF instrument and OpenSwath for data analysis[37].

Furthermore, DIA-NN, Skyline, and Spectronaut correctly yield 25% missingness for most *E. coli* proteins, while OpenSwath only reaches this distinction if TRIC is applied for FDR filtering (Supplementary Fig. 6). The nature of the spectral library only had a negligible impact in this regard in most cases. An exception to this rule is DIA-NN combined with the EncyclopeDIA-refined in silico predicted PROSIT spectral library, which in comparison to the other DIA-NN-library combinations showed a higher number of reported *E. coli* proteins for those samples that effectively did not contain any *E. coli* proteins.

At the precursor level, however, as opposed to the protein level, OpenSwath resembles the other software suites in terms of missingness, with most *E. coli* proteins showing a missingness of at least 25% (Supplementary Fig. 7a). This implies that the differing behavior of OpenSwath at the protein level may, at least in part, be due to the protein summarization it performs.

Furthermore, we assessed how the missingness within each sample correlates with the sample mean of protein intensities. While for DIA-NN and Spectronaut this correlation is positive showing a separation of the spike-in conditions by

sample mean of protein intensities, it is negative for Skyline and OpenSwath with neither of both showing such a separation (Fig. 3b).

We hypothesize that the counter-intuitive positive correlation between protein intensity and missingness, as in the case of DIA-NN and Spectronaut, may be due to sample-dependent detection thresholds[38]. In other words, if the intensity of a protein lies below such a threshold, it is not included into the calculation of the sample mean of the protein intensities, thus, increasing the weight of proteins with higher intensities. This, in turn, increases the sample mean of protein intensities. Interestingly, while at the protein level DIA-NN showed a positive correlation between missingness and sample mean, this correlation turns negative at the precursor level (Supplementary Fig. 7b).

The implications of these findings are far-reaching and should be taken into consideration when planning studies, as in practically all proteomics experiments missing values are an issue that needs to be addressed. To our knowledge, detection of true missingness and false-positive quantitation is rarely investigated in benchmarking studies. Our dataset offers a well-suited platform to investigate (and possibly optimize) these aspects for future toolsets.

**Data analysis scheme and performance measures.** Although complex in its own realm, protein and peptide identification and quantitation from LC-MS/MS data are only the beginning of the complete analysis of a multi-sample, quantitative proteomics experiment. Subsequent steps typically include sparsity reduction, normalization, and, ultimately, statistical assessment of differential protein abundance. For each of those steps different algorithms exist, yielding a variety of possible combinations.

To investigate the performance of the analysis methods in different possible combinations, we jointly assessed commonly used approaches for sparsity reduction, normalization, and

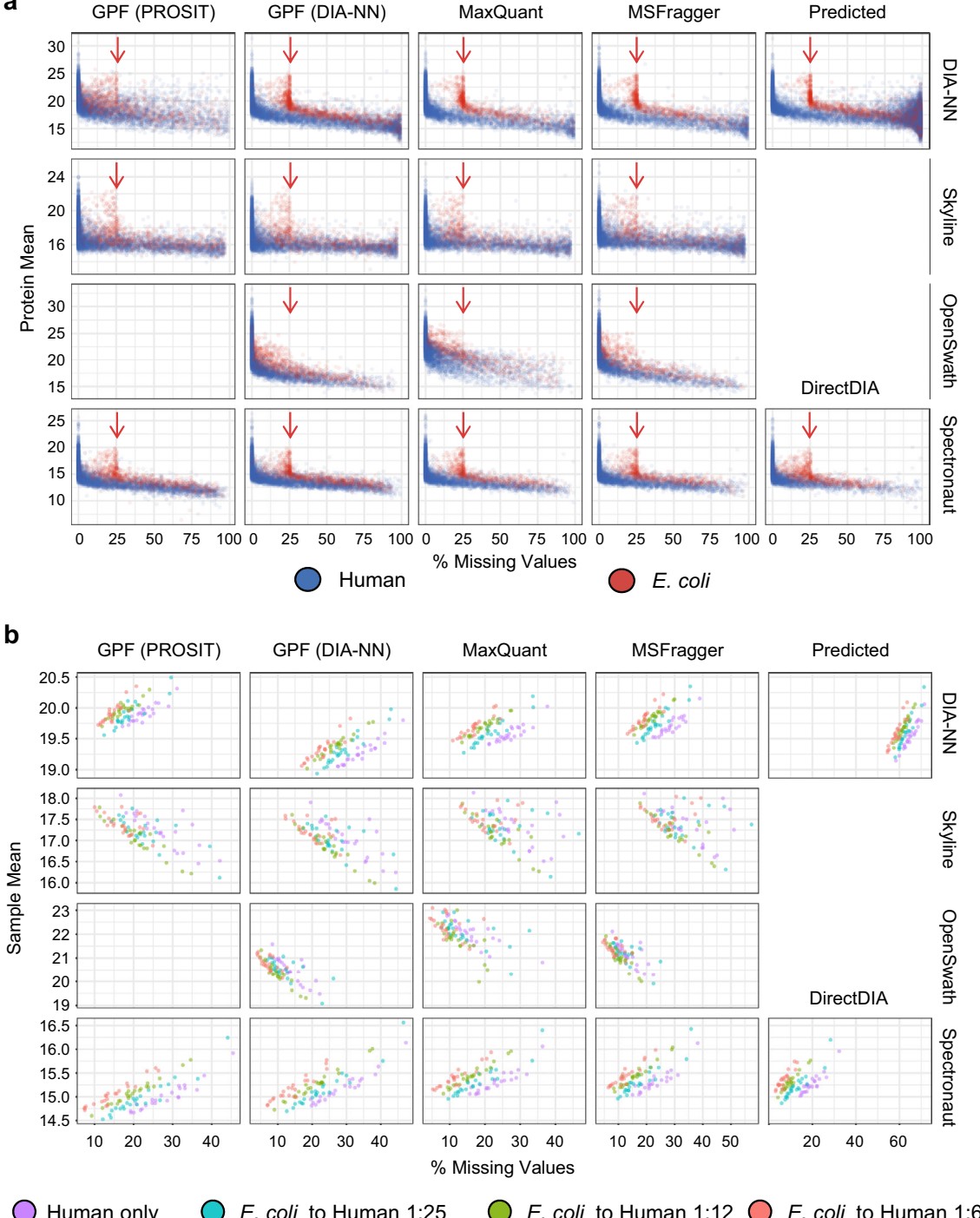

**Fig. 3 Missing value characteristics and correlations at the protein level. a** The way small intensities are handled mainly depends on the employed DIA software. Means of log2 intensities of identified human (blue) and *E. coli* (red) proteins plotted against the percentage of missing values in the respective protein. *E. coli* proteins are not physically present in 25% of samples (indicated by the red arrow). **b** The correlation between the missingness within samples and the sample mean over all proteins of these samples varies with the employed DIA software. Sample means are plotted against the percentage of missing values in the respective sample.

different statistical tests. For sparsity reduction we applied: (a) no sparsity reduction (NoSR), (b) requiring >66% values per protein (SR66), and (c) requiring >90% values per protein (SR90). Four different methods were then applied to investigate the effect of normalization: (a) unnormalized, (b) quantile normalization (QN), (c) tail-robust quantile normalization (TRQN), and (d) median normalization. Finally, we used the following seven statistical tests to probe for differentially abundant proteins:

Student's *t*-test, Welch's *t*-test, generalized linear model with Gamma family and log link (GLMgamma), linear models for microarray data (limma)[39], Wilcoxon–Mann–Whitney test (Wilcoxon), significance analysis of microarrays (SAM)[40], and reproducibility-optimized test statistic (ROTS)[22,41].

To systematically evaluate the performance of each of the above mentioned parameters, we focused on a sub-dataset, representing the two lowest *E. coli* spike-in conditions. We used bootstrapping

to quantify the uncertainty of the observed assessment score and to investigate the effect of sample size on the overall ability of the data analysis workflow to detect differentially abundant proteins. To this end, we randomly drew (with replacement) from the set of samples of the two lowest *E. coli* spike-in conditions to receive group sizes of three to 23 samples. Although bootstrapping is a well-established technique, it is known that it can introduce a bias for small sample sizes (i.e., in non-asymptotic settings). For a fair comparison of performances, it is most important that this potential bias is shared among the different statistical tests. This is, indeed, the case for the exemplarily chosen samples sizes 3, 6, 13, and 23 (Supplementary Fig. 20). To each bootstrap dataset, we applied all combinations of the aforementioned sparsity reductions, normalizations, and statistical testing options to determine differentially abundant proteins.

To objectively compare the different data analysis workflows, we introduced a measure of performance for detecting differentially abundant proteins. The experimental design with the known *E. coli* spike-in conditions provides us with ground truth information based on which we can assess true positives (*E. coli* proteins, which are determined to be significantly differentially abundant between the two spike-in conditions), and false positives (human proteins determined to be significantly differentially abundant between the two spike-in conditions), false negatives (*E. coli* proteins determined to be non-significantly differentially abundant between the two spike-in levels), and true negatives (human proteins determined to be non-significantly differentially abundant between the two spike-in conditions). For each protein, we can then plot the true-positive rate against the false-positive rate to obtain a receiver operating characteristic (ROC) curve.

As a measure of the ability of each workflow to detect differentially abundant proteins, we determined the area under the ROC curve. We use the partial area under the curve (pAUC) for all analyses, as it captures the area of a low false-positive rate (FPR, 1-specificity), which in practice is the most relevant one (Fig. 4a). While the AUC can be interpreted as the average true-positive rate (TPR, sensitivity) across the whole range of specificities, pAUCs correspond to the average TPR over a relevant (often low) FPR range only[21,42]. Here, for the calculation of pAUC we focus on FPRs below 10%. To enhance comparability, the maximum pAUC value that can be reached this way is scaled up to 100%.

Although the fold-changes of the spiked-in *E. coli* proteins are known through our study design, it is unknown which human and *E. coli* proteins were actually present in the biological sample in the first place. Since the TPR and FPR calculations strongly depend on the definition of the set of proteins present, we calculated them based on three different protein lists. This allows us to evaluate the robustness of the outcomes, while ensuring that no software or library is favored.

The proteins, which are present in the DIA analysis workflow dataset, a given bootstrap dataset has been drawn from, are collectively referred to as 'DIA Workflow' proteins (Supplementary Fig. 10 and 11). The list of proteins which were identified in at least one of the DIA analysis workflows is referred to as 'Combined' (11,533 Human proteins, 2125 *E. coli* proteins). The list of proteins which were identified in >80% (at least 14 out of 17) of the DIA analysis workflows is referred to as 'Intersection' (4512 human proteins, 740 *E. coli* proteins), and represents a list of proteins common to most DIA analysis workflows.

A summary of those three lists is given in Supplementary Fig. 9. In contrast to the other two reference protein lists, the 'DIA Workflow' list is unique for each DIA analysis workflow. The 'Combined' reference protein list, due to its larger size, likely contains a higher number of true-positive and false-positive protein identifications and a lower number of true-negative and false-negative protein identifications as well as a lower quantification quality compared to the other reference protein lists. For the 'Intersection' reference protein list the opposite is the case. Here, quantification quality refers to the fact that proteins which are difficult to identify are also more difficult to quantify and thus tend to reduce the performance of the analysis. The maximum TPR that can be reached is 1 for the 'DIA Workflow' and 'Intersection' reference protein list, while it is DIA analysis workflow-dependent for the 'Combined' reference protein list, where the maximum TPR is given by the number of identified proteins identified by the respective DIA analysis workflow over the number of proteins in the 'Combined' reference protein list.

**Data analysis results**. The impact of the workflow steps and their choices on the prediction performance quantified by calculating the pAUC decreases in the following order: (1) DIA analysis workflows and reference protein lists, (2) sparsity reduction, (3) normalization and statistical tests.

Both the choice of a DIA analysis workflow and an appropriate reference protein list strongly impact the outcome of our workflow comparisons. Figure 4c shows the performance of each DIA analysis workflow separated by reference protein list.

For some DIA analysis workflows, the performance between different reference protein lists differs drastically. For example, the DIA-NN predicted workflow detected the most unique proteins (Supplementary Figs. 10 and 11) and, thus, is likely to achieve a high sensitivity if the pAUC is calculated based on the 'Combined' reference protein list. However, as not all of the detected proteins by this specific workflow are reliable and only quantified in a few samples, this causes the worst of all DIA workflows performances when calculating the pAUC based on the 'DIA workflow' reference protein list. Also, as the 'match-between-runs' function had not yet been included in the DIA-NN version we used for this study, the obtained results should be revisited for newer DIA-NN versions.

The strong dependence of the performance of each DIA analysis software suite on the spectral library with which it is combined, and on the protein list against which it was benchmarked, is demonstrated as follows:

We assess the 'within workflow' performance by using the distinct 'DIA Workflow' protein lists to measure the prediction performance of differentially abundant proteins. We find that Spectronaut's 'DirectDIA' performs best, while DIA-NN, Skyline and Spectronaut all perform well using the more classical DDA spectral libraries generated by MaxQuant and MSFragger. Combining OpenSwath with the MSFragger-based spectral library leads to a better prediction performance than combining it with the MaxQuant spectral library. The opposite is the case for Skyline. Overall, the GPF-refined libraries show an inferior performance, except for the refined DIA-NN spectral library in combination with OpenSwath.

The 'overall sensitivity' performance is assessed by including proteins of all workflows into the reference protein list. When calculating the pAUC based on this 'Combined' reference protein list, the GPF-refined libraries, but not the in silico predicted DIA-NN unrefined library, perform well for DIA-NN and OpenSwath workflows. These libraries do not, however, perform as well for Spectronaut. Skyline performs better with the refined PROSIT spectral library as compared to the refined DIA-NN spectral library for this specific reference protein list. Also, the DDA-based spectral libraries perform worse than in the case of the 'DIA Workflow' protein list.

Using the 'Intersection' reference protein list, on average, DIA-NN performs slightly better than the other software solutions. The refined DIA-NN library in combination with

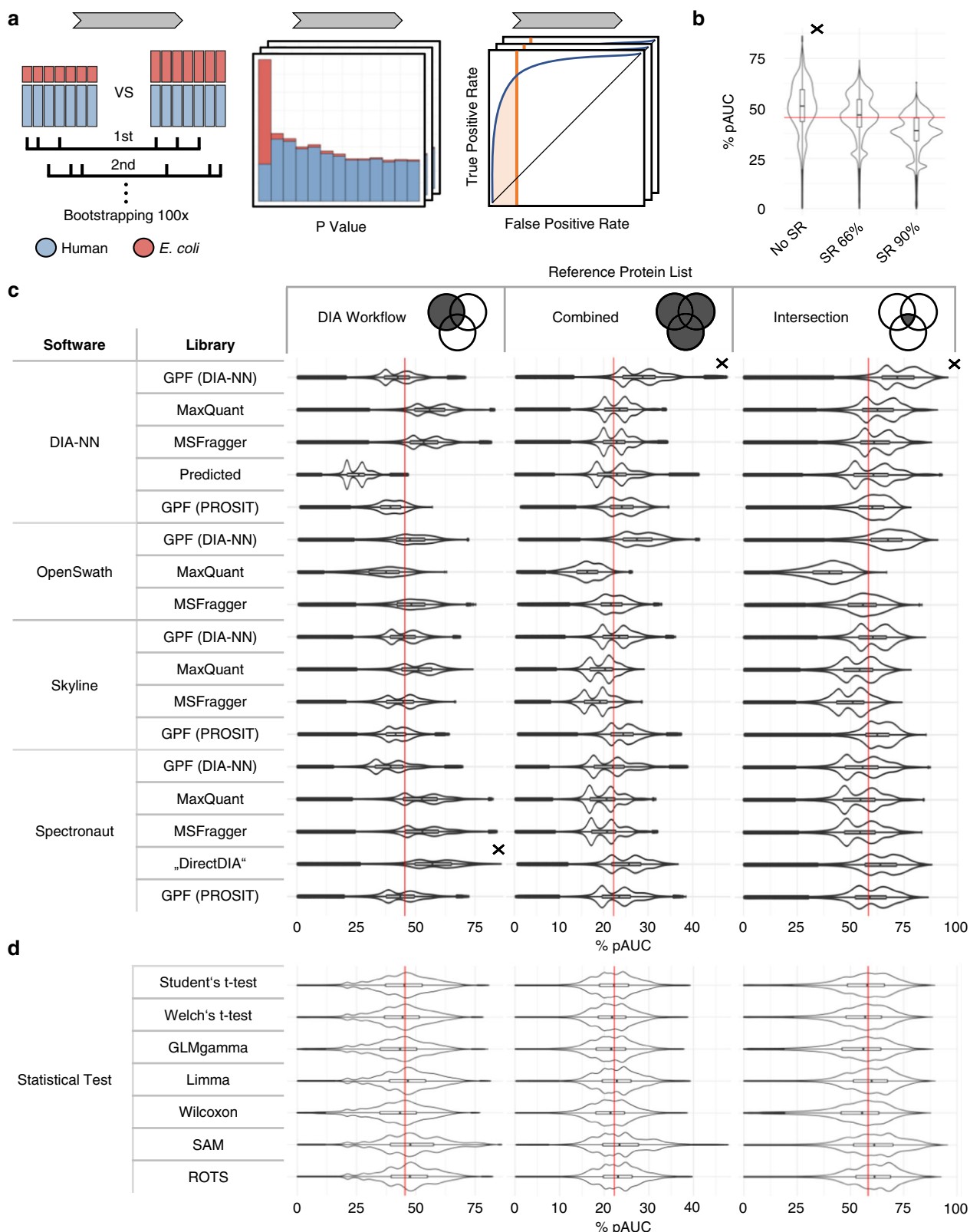

DIA-NN and OpenSwath, but not with Skyline and Spectronaut, leads to a good prediction performance against the 'Intersection' protein list.

These data highlight the strengths, but also the limitations of any spectral library-DIA software combination. While some combinations lead to a high number of reported proteins, others will result in a lower number but at the same time in a higher

pAUC, or give more accurate results in terms of estimated vs. true log2 fold-changes (Supplementary Fig. 12).

Furthermore, as opposed to the choice of a statistical test (and a normalization method) (Fig. 4d), the choice of a DIA analysis workflows (Fig. 4c) has a much bigger impact on the pAUC that can be obtained. The multimodal distribution of pAUC, which is particularly prominent for DIA-NN, Skyline, and Spectronaut, is

**Fig. 4 Statistical analysis of benchmark dataset. a** Workflow schematic: for the generation of bootstrap datasets, random samples were drawn with replacement from samples of spike-in conditions 1:25 and 1:12 mimicking two groups containing differentially abundant proteins, here represented by all *E. coli* proteins. The *p*-values acquired after data preprocessing and statistical analysis were used to build receiver operating characteristic (ROC) curves. The partial area under the curve (pAUC) was used as a measure of prediction performance. **b** pAUC distribution for the different sparsity reduction options (as measured against 'DIA Workflow' protein list). **c** pAUC for the different DIA analysis workflows as measured against the three reference protein lists. **d** pAUC distributions for the statistical tests. All seven statistical tests were two-sided and not adjusted for multiple testing. 'DIA Workflow' describes the performance against the proteins present in the given DIA workflow only, 'Combined' describes the performance against proteins identified at least by one of all DIA analysis workflows. 'Intersection' describes the performance against proteins which were found in >80% (in at least 14 of 17) of the DIA analysis workflows. For each reference protein list, the respective median of all pAUC values is indicated by a red line, and the best performing option with a cross. **b–d** are based on $n = 2100$ bootstrap datasets which have been generated by drawing with replacement from data of $n = 23$ biologically independent samples of spike-in conditions 1:12 and 1:25, respectively. The sample size of these bootstrap datasets ranged from 3 to 23 samples, which due to drawing with replacement can appear multiple times. For **c** that comes to a total of $n = 2100 * 17$ DIA workflows $* 4$ normalizations $* 7$ statistical tests $= 999600$ data points per sparsity reduction setting, for **c** to a total of $n = 2100 * 3$ sparsity reductions $* 7$ statistical tests $= 176400$ per library-software combination, and for **d** to a total of $n = 2100 * 17$ DIA workflows $* 3$ sparsity reductions $* 4$ normalizations $= 428400$ per statistical test setting. The boxplots show median (center line), interquartile range (IQR, extending from the first to the third quartile) (box), and $1.5 * IQR$ (whiskers).

due to the three included sparsity reduction options. For OpenSwath, on the other hand, this multimodal distribution is less pronounced. Supplementary Fig. 19 summarizes the ranked performance of DIA analysis workflows for different choices of sparsity reduction, normalization, and statistical test.

Third in line in terms of impact on the prediction performance, after the choice of DIA analysis workflow and reference protein list, is the extend of sparsity reduction that is selected.

The pAUC values shown in Fig. 4b are calculated based on the 'DIA Workflow' protein list. The highest pAUC values are achieved if no sparsity reduction is performed, while stricter criteria for missing values resulting in the removal of proteins lead to a decrease in pAUC. While the removal of proteins via sparsity reduction can lead to a situation in which the maximum sensitivity cannot be reached, the same can happen if the reference protein list, based on which TPR (and FPR) is calculated, and with it the number of *E. coli* proteins, is larger than the list of proteins, for which statistical results are available. Furthermore, we observed a steep initial increase in the ROC curve for SR90, after which a plateau is reached. Differences in the pAUC are then solely based on the height of this plateau, which itself depends on the number of quantified *E. coli* proteins in a given dataset. If testing for differential abundance of a protein returned a missing value, the *p*-value for this comparison was set to one. As human proteins are overrepresented in our benchmark dataset, this might lead to a bias when performing sparsity reduction, limiting inter-comparability of sparsity reduction levels.

We found that virtually all DIA software-spectral library combinations do not benefit from normalization and perform best with unnormalized data (Supplementary Fig. 13). All normalization methods included in this study normalize by distribution and, thus, act under the assumption of a symmetric (balanced) differential expression, i.e., that the number of up- and downregulated proteins is equal[43]. In this benchmark dataset, however, the differentially expressed proteins solely change in one direction. Thus, we hypothesize that the observed decline of performance when normalization is performed could, at least partly, be an artifact of the study design. This highlights the importance of correctly interpreting benchmark results. Also in a practical setting, e.g., for knock-down models or during proliferation but also in clinical settings, it is possible that the number of up- and down-regulated analytes is unbalanced[25]. As a way of conducting normalization on an unbalanced dataset, normalization can be performed on merely a subset of all measured analytes, which consists of invariant analytes only. This approach has already been embraced in transcriptomic settings[44].

Furthermore, the impact of human proteins on the normalization outcome is higher than of *E. coli* proteins, due to the larger number of human proteins being present in the samples. As a result, the distribution of the human proteins is comparable across the normalized samples, while this is not the case for the *E. coli* proteins. This might lead to a bias in the identification of differentially abundant *E. coli* proteins.

Additionally, all employed normalization methods assume that the relative abundance of proteins within one sample can be used to normalize all proteins. This, however, cannot be assumed for this dataset as *E. coli* and human proteins were pipetted separately, which leads to changes in the protein abundance ranks between samples (which are assumed to be stable by the normalization methods). This highlights the need to employ special strategies to evaluate normalization strategies in future benchmark studies. A dilution series of the same samples being measured with different injection amounts may be more suitable to investigate normalization methods.

Finally, we evaluated the prediction performance of statistical tests for two-group comparisons, that have previously been used in proteomics data analysis (Fig. 4d), again for all three reference protein lists. In general, the non-parametric permutation-based test SAM consistently performs best for all DIA analysis workflows followed by the other non-parametric permutation-based test ROTS and limma (Supplementary Fig. 14). However, the superiority of SAM over ROTS might be due to the set hyperparameters, as the SAM statistic can be derived from the more general ROTS statistic[42,45,46]. Interestingly, in the study of Pursiheimo et al. SAM did not perform well, while the good performance of ROTS is highlighted[20]. Although the good performance of non-parametric methods has been described previously[42,45] the simple non-parametric Wilcoxon–Mann–Whitney test, together with GLMgamma, performs worst in our setting, suggesting that the good performance of SAM and ROTS stems from their inherent permutation step.

**Impact of data characteristics on prediction performance.** We investigated the connection between data properties of the bootstrap datasets and statistical prediction performance. As we identified DIA-NN in combination with the in silico predicted GPF-refined spectral library as an overall well-performing DIA analysis workflow, we further investigated which data properties (Supplementary Fig. 15) correlate with benchmarking performance measures (Supplementary Fig. 16).

In general, sample variance, kurtosis, skewness, and the ratio of variances between two spike-in conditions only slightly influence the performance of statistical tests to detect differentially

abundant proteins, and the correlation behavior of the data characteristics differs between the DIA analysis workflows (Supplementary Fig. 17).

As we included different sample sizes during bootstrapping to mimic limited replicate availability in practice, we were also able to investigate the performance of the different DIA analysis workflows for different sample sizes (Supplementary Fig. 18). We observed a moderate positive correlation between pAUC values and sample size for all DIA analysis workflows (shown in Supplementary Fig. 16 at the example of DIA-NN in combination with the in silico predicted GPF-refined spectral library).

Overall, SAM performed best for all sample sizes over all workflows, with ROTS and limma achieving a similar performance for small sample sizes ($n < 5$). Van Ooijen et al., who compared different statistical tools to detect differentially abundant proteome features, also found limma to perform well for small sample sizes[47].

In conclusion, with this comprehensive benchmark study in which we assessed multiple processing options simultaneously, we strive to support the proteomics community by providing deeper insights into the interaction between spectral libraries, DIA software suites, data preprocessing, and statistical testing for differentially abundant proteins.

From the four DIA software suites we investigated we found that DIA-NN, Skyline, and Spectronaut robustly avoided the false detection of *E. coli* proteins, which are truly absent in human-only samples. However, the same can be achieved by using the TRIC tool in the OpenSwath Galaxy workflow. This is highly relevant for studies inferring biological relevance from missing values, especially in a clinical context.

Naturally, the amount of missing values also influences the effect of sparsity reduction. This effect is smaller for OpenSwath as compared to DIA-NN, Skyline, and Spectronaut.

By conducting multiple analysis workflows on bootstrap samples derived from 17 DIA software-library combinations, we found that the choice of DIA analysis workflows and reference protein lists had the highest impact on the prediction performance for differentially abundant proteins, followed by the choice of sparsity reduction, while the choice of normalization and statistical tests had only a minor impact.

The ability of the DIA software-library combinations to detect differentially abundant proteins varied with the reference protein list that was used to calculate the true-positive and false-positive rate. When all proteins were included appearing in at least one DIA software-library combination, DIA-NN in combination with the in silico predicted DIA-NN spectral library (GPF-refined by DIA-NN) performed best. The same is true when considering the 'Intersection' reference protein list (proteins identified in 80% of all DIA analysis workflows). If only those proteins are taken into account, which were found in the dataset of the respective DIA software-library combination, Spectronaut's 'DirectDIA' excelled. This highlights the importance of spectral library generation and the quality of the resulting spectral libraries. For in silico prediction of spectral libraries (and their refinement on LC-MS/MS measurements), which is currently gaining momentum, the choice of library prediction algorithm, possible LC-MS/MS refinement, and the actual DIA analysis software lead to even more complex combinatorial workflows.

In general, the depth of proteome coverage can be increased by using GPF-refined libraries. While this leads to an increase in low quality identifications or quantitations, we see that the positive effects of GPF-refined libraries on the quantitation of the 'Intersection' reference protein list outweigh the negative effects.

The pAUC strongly decreased with stricter sparsity reduction options, which is due to the increased removal of *E. coli* proteins containing missing values, which leads to a decrease in sensitivity.

The minor impact of the choice of normalization highlights the quality of internal protein inference and summarization algorithms for all tools, especially for DIA-NN and Spectronaut.

As for statistical testing, the non-parametric permutation-based statistical tests SAM consistently performed best, followed by ROTS and limma. However, as the Wilcoxon–Mann–Whitney test as a simple non-parametric test shows an inferior performance, this implies that the good performance of SAM and ROTS is due to these algorithms performing permutations. Looking at the parametric statistical tests, limma performed best compared to other parametric statistical tests, especially for very low sample sizes.

In the future, we strive to further exploit the data characteristics information we acquired for each bootstrap dataset, by looking for interactions between those data characteristics and algorithm performances in order to provide algorithm recommendations for researchers with datasets of certain characteristics.

In summary, we found that the reliability and reproducibility of proteomics data analyses heavily depend on properly choosing and combining the options provided for each proteomics workflow step, as downstream analyses rely on certain assumptions about data characteristics, e.g., regarding missing values, which are themselves heavily influenced by the choice of DIA software and spectral libraries.

We encourage others to assess their own approaches and workflows – specifically those used in clinical settings – using our dataset as it mimics a realistic biomedical setting with its inherent heterogeneous background.

## Methods

**Sample preparation**. The study has been approved by the Ethics Board of the University Medical Center Freiburg (approval 280/18) and written patient consent was obtained before inclusion. The study design and conduct complied with all relevant regulations regarding the use of human study participants and was conducted in accordance with the criteria set by the Declaration of Helsinki. Histologically non-infiltrated lymph nodes from patients with acinary prostate cancer were collected (with consent) as sentinel samples and preserved as FFPE tissue. Consecutive slices of 10 μm thickness were deparaffinized, stained, and macro-dissected to acquire 0.5–1 mm³ of lymph node tissue per patient. Subsequently, antigen retrieval was performed in 4% (v/v) SDS, 100 mM HEPES pH 8.0, with samples being sonicated using a Bioruptor device for 10 cycles (40 s/20 s, high intensity), heated to 95 °C for 1 h, and sonicated again.

*E. coli* K12 bacteria were provided by Christoph Schell (University Medical Center, Freiburg) as cell pellets. *E. coli* samples were lysed in 4% SDS in 100 mM HEPES pH 8 and heated to 95 °C for 10 min and subsequently sonicated using a Bioruptor for 15 cycles (40 s/20 s, high intensity).

All samples were centrifuged at $15{,}000 \times g$ for 10 min at room temperature. Only the supernatant was used for MS sample preparation.

FFPE tissue samples and *E. coli* samples were reduced at 95 °C for 10 min using 5 mM TCEP. Samples were alkylated for 20 min at room temperature in the dark using 10 mM iodoacetamide. Samples were prepared for MS analysis using micro S-TRAP columns (PROTIFY) according to manufacturer's instructions[48]. In brief, protein concentration of samples was determined using a BCA assay (Thermo) and 25 μg of protein was loaded onto separate columns after acidifying (with phosphoric acid) and diluting the sample (1:6 v/v to sample volume corresponding to 25 μg) with loading buffer (100 mM TEAB pH 7.1 in 90% MeOH). The samples were then washed four times using loading buffer. For digestion, a mix of trypsin and Lys-C (Promega, 1:20 w/w to sample protein amount) was used in 50 mM TEAB pH 8 and incubated at 47 °C for 1 h. Resulting peptides were eluted from the columns in three steps: (a) 50 mM TEAB pH 8, (b) 0.2% formic acid and (c) 50% acetonitrile, 0.2% formic acid. No further peptide purification was performed, and peptide content was measured using BCA assay (Thermo). Aliquots containing 5 μg peptide per sample were dried and stored at −80 °C until measurement.

For spectral library generation, a masterpool sample with an *E. coli* to lymph node peptide ratio of 1:15 was generated by combining peptides from 12 randomly chosen samples (three from each spike-in condition). For DDA library generation, the master mix sample was pre-fractionated using offline high-pH prefractionation as described previously[49]. In brief, 45 μg of master mix were taken and loaded onto an XBridge Peptide BEH column (Waters, 150 × 1 mm) coupled to an Agilent 1100 HPLC system and separated using a 45 min linear gradient from 4 to 30% acetonitrile in 10 mM ammonium formate adjusted to pH 10. Of the 24 fractions collected in total, the 20 fractions, which contained peptides according to the chromatogram, were pooled using the following scheme: 1 + 11, 2 + 12, etc., resulting in ten pooled samples.

**LC-MS/MS measurements**. All LC-MS/MS runs were acquired using an Orbitrap Eclipse Mass Spectrometer (Thermo) coupled to an Easy nLC 1200 (Thermo). Self-fritted precolumns (Frit Kit, Next Advance) with 100 µm ID were self-packed with 3 µm C18 AQ (Dr. Maisch) beads to a length of 2 cm. A 75 µm ID Picofrit column (New Objective) was self-packed with 1.9 µm C18 AQ (Dr. Maisch) beads to a length of 20 cm as previously described[50]. For every injection, 500 ng of peptides were used. iRT peptides (Biognosys) were added to a final quantity of 50 fmol/injection. Buffer A consisted of 0.1% formic acid, buffer B consisted of 80% acetonitrile in 0.1% formic acid. All samples were separated using a 70 min linear gradient from 5 to 31% B followed by a 5 min linear gradient from 31 to 44% buffer B. For the data acquisition of the dilution series the mass spectrometer was operated in DIA mode and the standard parameters from the staggered DIA method editor node were used. Briefly, a survey scan (60k resolution) from 390 to 1010 m/z was followed by MS2 scans (15k resolution) with 8 m/z isolation width covering 400 m/z to 1000 m/z. A second survey scan was followed by MS2 scans with an offset of 4 m/z as compared to the first cycle. For MS2 scans, peptides were fragmented using HCD and stepped collision energy 30 (5%), and maximum injection time was set to 22 ms. The data were recorded in centroid mode. For GPF measurements, the masterpool sample was repeatedly measured. A tSIM scan with an isolation width of 110 m/z was followed by MS2 scans with 4 m/z isolation width over 100 m/z. A second tSIM scan with 110 m/z was followed by MS2 scans with an offset of 2 m/z as compared to the first cycle. A total of six measurements were performed to cover a scan range from 400 to 1000 m/z. For data-dependent acquisition measurements, the masterpool sample was pre-fractionated offline prior to LC-MS/MS measurement as described previously (see above). A survey scan of 120k ranging from 390 m/z to 1010 m/z was recorded. Following the survey scan, a Top 15 method was employed. MS2 scans were recorded at 15k resolution with the isolation window set to 1.6 m/z and maximum injection time set to 60 ms. Pre-fractionated samples were measured in duplicates. DDA data integrity was validated using PTXQC[51]. All acquisition methods are accessible by downloading the raw data and extracting the instrument method from the raw files.

To generate experiment-specific spectral libraries, we performed GPF on a master mix, which represents an average spike-in concentration of *E. coli* to human peptides of 1:15. Using DIA-NN to refine a combined *E. coli* and human in silico predicted DIA-NN spectral library, we generated a spectral library containing 84016 precursor entries mapping to 10459 proteins. Using an in silico predicted PROSIT spectral library refined by EncyclopeDIA, we generated a spectral library containing 45445 precursors mapping to 8472 proteins[18].

We also pre-fractionated a master mix to obtain samples for in-depth DDA library generation[29]. By applying Fragpipe to the resulting DDA files we generated a spectral library containing 81409 precursors mapping to 7781 proteins. We also used MaxQuant to build a DDA-based spectral library containing 51260 precursors mapping to 7382 proteins.

**Spectral library generation**. For all spectral libraries, a reviewed human and *E. coli* K12 FASTA (one entry per gene) were downloaded from Uniprot on Nov 22nd 2020[52]. The GPF-refined PROSIT[53] spectral library was generated as described previously[18]. In brief, EncyclopeDIA[35] (0.9.5) was used to generate PROSIT input csv files. PROSIT (2019 iRT prediction model) was used to predict spectra and retention times, which were reimported into EncyclopeDIA. Destaggered GPF mzml files were then used to generate a GPF-refined library, which was exported in tabular format. The GPF-refined DIA-NN spectral library was predicted and refined using DIA-NN. In brief, DIA-NN was provided with a combined FASTA protein database (human + *E. coli*) as input and neural networks were used to generate spectra and retention times for the appropriate mass range of 390–1010 m/z. N-terminal methionine excision was enabled. Carbamidomethylation of cysteine was activated as fixed modification. The GPF mzml files were then used (with the same settings as described in the spectral library prediction step) to generate a GPF-refined library, which was exported in tabular format. For the DIA-NN in silico predicted library, the refinement step was skipped and the unrefined in silico predicted library (of all FASTA entries) was directly used for DIA analysis. The MaxQuant DDA library was generated using MaxQuant (1.6.14.0) searching the DDA files resulting from prefractionation directly as raw files. Carbamidomethylation of cysteine was set as fixed modification, whereas oxidation of methionine and acetylation of the protein N-terminus were set as variable modification. A first search with a precursor mass tolerance of 20 ppm was performed, followed by the main search of the recalibrated data with 4.5 ppm mass tolerance. The false discovery rate was set to 1% on peptide-spectrum-match and protein level. The MaxQuant output was imported into Spectronaut, converted to library format and exported in tabular format. The MSFragger DDA library was generated using MSFragger (3.2) in the Fragpipe GUI (14.0) in conjunction with easyPQP (0.1.25), following conversion of DDA raw files to mzXML format. Precursor mass tolerance was set to 20 ppm and mass calibration as well as parameter optimization was enabled. Carbamidomethylation of cysteine was set as fixed modification, whereas oxidation of methionine and acetylation of the protein N-terminus were set as variable modification. MSFragger output was converted to tabular format using DIA-NN. Raw files were destaggered and converted to mzML or mzXML format using MSConvert in conjunction with ProteoWizard (3.0.20315)[54] or demultiplexed and converted to htrms format using Spectronaut HTRMS converter (14.0).

**DIA data acquisition and processing**. DIA-NN (1.7.12) was used with recommended settings. Mass ranges were set appropriately for the search space (MS1: 390 m/z to 1010 m/z; MS2: 150 m/z to 1500 m/z) and RT profiling was activated. For the in silico predicted library search, the reduced memory option was additionally activated. Protein and precursor FDR was set to 1%. All DIA-NN computations were performed on an Intel(R) Xeon(R) Gold 6246 CPU.

Skyline[55] (64 Bit) (20.2.0.343) analyses were performed as described in the Skyline tutorials 'Analysis of DIA/SWATH data' and 'Advanced Peak Picking Models'. In brief, the 'Import Peptide Search' daemon was used to import spectral libraries and implement the iRT retention time predictor[30]. The allowed deviation of measured retention times versus predicted retention times was set to 3 min. Mass accuracy was set to 10 ppm. An mProphet model was trained not including MS1 information and results were filtered based on the q-value given by the mProphet model (1% precursor FDR). Results were then imported into MSstats (3.21.3), where protein summarization was performed using the 'MSstats' protein summarization method[56], which was adjusted (by modifying MSstats::preProcessIntensities() which is called by MSstats::MSstatsPrepareForDataProcess()) such that intensities below one were not set to 1 as would be the default. Also, for better comparison with the other DIA workflows, no imputation and normalization has been performed.

The OpenSwath[34] Workflows (2.6) were used in Galaxy with the default settings except for minor adjustments as previously described[57,58]. Briefly, the mass accuracy on the MS1 and MS2 level was set to 10 ppm. For iRT peptide extraction, 20 ppm was used, and a minimum of seven iRT peptides was requested. Target-decoy scoring was performed using PyProphet (2.1.4.2) in Galaxy with the 'XGBoost' classifier for semi-supervised learning including MS1 as well as MS2 information[59]. Identification results were filtered based on a peptide and protein FDR of 1% using PyProphet.

To assess the impact of an alternative FDR filtering and feature alignment approach in the OpenSwath analysis (Supplementary Fig. 6), TRIC (0.11.0) was used with two different max_fdr_quality settings, of 1 and 5%, respectively[36]. The alignment score was set to 0.0005.

Spectronaut's (14.0) 'import' function was used for converting tabular libraries into Spectronaut format. Before import, the retention times of the PROSIT-EncyclopeDIA and the DIA-NN libraries were converted to minutes and a linear model was used to convert retention times to iRT values.

Prior to performing data analysis on the protein intensity datasets derived from the 17 DIA workflow analyses, data were transformed to a common format, in which all proteins were annotated with their respective UniProtKB/Swiss-Prot entry names. For some DIA analysis workflows, some protein identifiers were composed of multiple protein names. Proteins were excluded if they were labeled both as human and as derived from *E. coli*. Proteins without reported quantitations were removed. For further analysis and visualization, the resulting protein intensities were log2-transformed. Peptide-level data were treated analogously. Due to corrupted recording of sample 28 of spike-in condition 1:6 it was not included in the analysis.

**Data analysis workflow**

*Bootstrapping*. To evaluate data analysis workflows—comprised of different combinations of sparsity reduction, normalization, and statistical tests options—in regard to their ability of identifying differentially abundant proteins in omics data, the two lowest human:*E. coli* spike-in ratios 1:25 and 1:12 have been used (Fig. 1). Bootstrap datasets were generated by randomly drawing (with replacement) a defined number of samples from each of the two spike-in conditions. We varied the group sizes for each spike-in condition from three to 23 samples and generated 100 bootstrap datasets for each of those group sizes, resulting in 2100 bootstrap datasets in total. To each of those bootstrap datasets all data analysis workflows have been applied.

*Sparsity reduction*. We included the following three sparsity reduction options: including all protein entries (NoSR), only those protein entries present in at least 66% (SR66), or 90% (SR90) of all samples.

*Normalization*. We included the following four normalization options: no normalization (unnormalized), tail-robust quantile normalization (TRQN)[38], quantile normalization (QN)[60,61], and median normalization (median).

*Statistical tests*. The following seven statistical tests were included in our analyses: Student's t-test (with equal variance assumption, t-test equal Var), Welch's t-test (with unequal variance assumption, t-test unequal Var), generalized linear model with Gamma family and log link (GLMgamma, applied after back-transformation of the log2-transformed intensities), linear models for microarray data (limma)[39], Wilcoxon-Mann–Whitney test (Wilcoxon), significance analysis of microarrays (SAM) (250 permutations used to estimate false discovery rates)[40], and reproducibility-optimized test statistic (ROTS)[22,41] (with 100 bootstrap and permutation resamplings and the largest top list size considered being 500. For run-time reasons this largest top list size is chosen to be smaller than recommended. Thus, a larger value of this setting is likely to improve the performance of ROTS). All conducted statistical tests were two-sided.

*Performance measures.* In total, we acquired performance information for 2100 bootstrap datasets × 17 DIA analysis workflows × 3 sparsity reduction options × 4 normalization options × 7 statistical test options = 2,998,800 cases. For each of those cases, we received for each protein of a bootstrap dataset a *p*-value as a result of a statistical test and the estimated log2-fold change (log2FC) between the two different *E. coli* to human spike-in ratios 1:25 and 1:12.

To evaluate which analysis workflow performed best in predicting the differentially abundant proteins, we used the partial area under the receiver operating characteristic (ROC) curve (pAUC). For the calculation of pAUC below a false-positive rate of 10% we used a modified version of the 'auc' function of the pROC R package[62]. For better comparability we set the maximal value that can be reached to 100%. If statistical tests returned a missing value for a given protein, the *p*-value of this protein was set to 1 in the respective analysis. We also calculated the sensitivity at a significance level of 0.05.

To quantify the preciseness of quantification, we calculated the root-mean-square error (RMSE) based on the estimated log2FC and the true log2FC, which is 0 for human proteins and 1.11 for *E. coli* proteins.

We calculated the performance measures for three reference protein lists in parallel: the 'Intersection' reference protein list with proteins appearing in >80% (at least 14 of 17) DIA analysis workflow datasets, the 'Combined' protein list with proteins appearing in at least one DIA analysis workflow dataset, and the 'DIA Workflow' protein list, which is specific for each DIA analysis workflow.

**Reporting summary**. Further information on research design is available in the Nature Research Reporting Summary linked to this article.

## Data availability

The raw data, libraries, analysis log files, and analysis output files data are available under restricted access for medical data protection reasons at the European Genome-phenome Archive (https://ega-archive.org) under the accession code EGAD00010002223. Access can be obtained via a data access agreement. The data access agreement for this dataset corresponds to the 'harmonised Data Access Agreement (hDAA) for Controlled Access Data' as brought forward by the 'European standardization framework for data integration and data-driven in silico models for personalized medicine—EU-STANDS4PM'. Please contact the corresponding author for access oliver.schilling@uniklinik-freiburg.de. Requests will be answered within 2 weeks.

Source data, including the processed protein and peptide intensity data as well as the benchmarking results and data characteristics, are provided with this paper at Zenodo (https://doi.org/10.5281/zenodo.6379087)[63].

## Code availability

The R code used for the statistical analyses is available at https://github.com/kreutz-lab/dia-benchmarking (https://doi.org/10.5281/zenodo.6371925)[64].

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

## Acknowledgements

The authors acknowledge support by the state of Baden-Württemberg through bwHPC and the German Research Foundation (DFG) through grant INST 35/1134-1 FUGG. The authors acknowledge the support of Björn Grüning from the Freiburg Galaxy Team, Bioinformatics, University of Freiburg (Germany) funded by the Collaborative Research Centre 992 Medical Epigenetics (DFG grant SFB 992/1 2012) and the German Federal Ministry of Education and Research BMBF grant 031 A538A de.NBI-RBC. O.S. acknowledges funding by the Deutsche Forschungsgemeinschaft (DFG, projects 446058856, 466359513, 444936968, 405351425, 431336276, 438496892 (SFB 1453 "NephGen"), 441891347 (SFB 1479 "OncoEscape"), 423813989 (GRK 2606 "ProtPath", 322977937 (GRK 2344 "MeInBio")), the ERA PerMed programme (BMBF, 01KU1916, 01KU1915A), the German-Israel Foundation (grant no. 1444), and the German Consortium for Translational Cancer Research (project Impro-Rec). This work was supported by the German Ministry of Education and Research by grant EA:Sys[FKZ031L0080] (C.K.) and by the Deutsche Forschungsgemeinschaft (DFG, German Research Foundation) under Germany's Excellence Strategy (CIBSS-EXC-2189-2100249960-390939984) (C.K. and E.B.). K.B. acknowledges funding by the Swiss canton of Grisons (protocol nr 628) and the Hans Groeber Foundation.

## Author contributions

Conceptualization: K.F., E.B., L.K., P.B., A.S., K.B., C.K., and O.S. Methodology: K.F., E.B., M.F., D.V., L.K., P.B., S.T.-B., and N.P. Validation: K.F., E.B., M.F., and N.P. Formal Analysis: K.F., E.B., M.F., D.V., P.B., S.T.B., C.K., O.S., and N.P. Resources: P.B., S.T.-B., A.S., K.B., C.K., and O.S. Data Curation: K.F., E.B., M.F., P.B., C.K., and O.S. Writing (Original Draft): K.F., E.B., L.K., K.B., C.K., and O.S. Writing (Review): K.F., E.B., K.B., C.K., and O.S. Visualizations: K.F., E.B. Project Administration: C.K., O.S. Funding acquisition: C.K., O.S.

## Funding

## Competing interests

The authors declare no competing interests.
