## [Peer Review File · Nature Communications]

REVIEWER COMMENTS

Reviewer #1 (Remarks to the Author):

The authors Fröhlich et al present an interesting approach to benchmarking using bootstrapping drawn from a large dataset. This enables the authors to better estimate parameters for which tool combinations work best in their background, and I think this is clever. In general, the work is well written, considered, and executed.

I found the abstract somewhat misleading and detracting from the work. The abstract currently presents the central advances of this work as benchmarking with a large dataset, the evaluation of 2M workflows, and some very narrow suggestions. To me, this manuscript is much more interesting than that! I recommend that the authors rewrite the abstract in the following ways:

a) It's easy to get a large dataset for testing and evaluation. The main innovation of this work is to develop a test dataset structured to generate bootstrapped smaller datasets, allowing for more robust statistical evaluation of different combinations of libraries, tools, etc. Rather than focus on the size of the dataset/number of evaluations, the abstract should discuss both the bootstrapping approach and how it helps enable more reliable assessments.

b) The take-home messages in the abstract seem far too specific, given the breadth of the evaluations. The abstract says that the specific combination of GPF/DIA-NN + DIA-NN + SAM produces reliable measurements, but realistically, all of the tested methods do that to one degree or another. Additionally, this statement ignores that GPF/Prosit + Skyline produced more quantified proteins than any of the other methods. To me, clearer take-home messages are (1) that GPF-based libraries improved the analysis of all of the software tools (and not just the ones they were designed for, e.g. EncyclopeDIA and Skyline), and (2) that non-parametric statistical testing consistently performed the best at detecting truly changing proteins.

c) I think the additional take-home message about missing values and false-positive quantification is interesting, but also likely due to the parameter settings for how the authors ran the different software tools (see below). I recommend dulling this message to not discuss performance of specific tools, but rather indicate that it is an important consideration or source of variability.

I recommend the authors consider the following additional points:

1) Page 8 line 280: The authors discuss how DIA-NN and Spectronaut report 0s for missing values, while Skyline and OpenSwath report background integration. These are just default choices for each

software, and those choices should be made on a dataset-by-dataset (or question-by-question) basis. For example, you can adjust OpenSwath's reporting for background integration by adjusting the `--max_fdr_quality` setting in TRIC. In Skyline, researchers typically filter out background signals based on other features, such as precursor isotope dot product or spectral correlation to the library. Missing values are an important point to discuss, but I suspect that all of these software tools have "toggles" to turn reporting of background integration on or off based on the researcher's use cases.

2) Page 10 line 330: The authors discuss using bootstrapping with replacement. I understand the reasoning, but the authors should discuss why they chose to do this and how this could be dangerous, since drawing duplicate samples is possible and may affect the success of downstream tools (e.g. this might help explain in part why non-parametric stats perform so much better than parametric stats, which assume an appropriate level of randomness in the data).

3) Page 11 line 356-361: The evaluation of "Combined" vs "Intersection" protein lists is confusing, especially since the "DIA-NN Predicted" workflow seems to detect a very different set of proteins than the rest of the tools (Supp Fig 5). The "Intersection" list seems sensible, but it's hard to reconcile the 11.5K proteins detected in "Combined" given that no single tool consistently detected more than 7K proteins. These lists should be clarified with respect to Fig 2.

4) Conclusions section: I think some discussion is needed of how this dataset of MS/MS runs is similar to more common Human experiments, but also illustrate the ways that it differs. In particular, the authors should indicate that these settings/results may not apply to other types of samples, such as biofluids, cell lines, etc due to the differences in complexity. Additionally, Human+Ecoli may not necessarily be representative of what Human samples are alone.

5) Methods section: The tools section regularly makes use of "Default" or "Recommended" settings. These settings should be clarified in the manuscript in case those recommendations change in the future.

Reviewer #2 (Remarks to the Author):

The authors present a novel report on a large benchmarking study comparing different Data Independent Acquisition (DIA) proteomics data analysis strategies using a cohort of patient samples. Authors used different amounts of Ecoli protein in patient samples to create populations of known concentration samples which will aid in assessment of effectiveness of different data analysis

strategies. DIA data analysis is heavily dependent on spectral library generation both empirically and in-silico, hence authors examined four broadly used data analysis software for both spectral library generation and DIA analysis to evaluate performance of each one. Using bootstrapping technique authors were able to evaluate how the sample size, normalization and all the subsequent statistical test are affecting the ability of the data analysis workflow to determine the correct ratio of Ecoli to human proteins.

DIA proteomics is quickly becoming a mainstream choice of analysis for increased identification and quantitation of proteins in a complex sample. This study has highlighted the need for benchmarking different DIA data analysis strategies especially using a patient cohort with inherent heterogeneity.

The authors have chosen appropriate methodologies to answer the main question of this study by choosing proper spiked in clinical samples for this study, performing all the sample preparation, LC/MS analysis and all the subsequent steps for data analysis and comparison of different spectral library generation workflows and analysis software packages. Authors have provided details of their experimental workflow and data analysis. Results are presented clearly and demonstrate the findings of authors in a logical manner.

Minor comments:

1) In lines 367-369 the following sentence is missing a word "In our study, the reference protein list based on which the ROC is generated represents an additional ?? that impacts the outcome of our comparisons."

2) Three out of the four data analysis software packages are using iRT peptides for retention time normalization. It would be beneficial if authors could comment about the effect of using iRT peptides for retention time normalization in the software packages which use it versus software package like DIA-NN which does not use iRT peptides. Do authors think that iRT peptides in any way, could affect identifications of peptides in a negative manner?

Reviewer #3 (Remarks to the Author):

In this study by Frohlich et al, the authors use protein digests derived from lymph nodes mixed with E.coli peptides to test a number of DIA software in combination with different spectral libraries and downstream data processing methods. The aim of the work is to determine the best DIA mass spectrometry and data analysis workflow for clinical samples.

The authors claim to determine a generalizable workflow for analyzing clinical samples by DIA mass spectrometry. However, as the authors claimed themselves, clinical samples are very heterogeneous in terms of protein expression levels and they can also be contaminated by e.g. cells from neighboring tissues in case of FFPE or by red blood cells in case of plasma). It is therefore difficult to

come up with an all-in-one solution for clinical sample analysis by DIA mass spectrometry. While a general guideline for DIA mass spectrometry could potentially be helpful, the experimental setup of the present study is very poorly explained and it is difficult to grasp what was actually done.

The work, as currently presented, is not suitable to target a broad readership as it is full of technical terms that are not properly explained (e.g. refinement of a spectral library, sparsity reduction).

Overall, the work might be relevant for a more specialized Journal but it is of limited novelty concerning both data processing and data analysis (all standard ways). The study is limited to testing a number of library/software/normalization/statistical tests combinations but lacks critical assessment of the results and data interpretation. Besides similar benchmark studies have already been published (e.g. <https://doi.org/10.1021/acs.jproteome.1c00490>).

Abstract:

It is not clear to me why the 92 lymph node samples result in 118 LC-MS/MS runs? I assume the library files were factored in?

Introduction:

- The intro could be more concise and would benefit from some (more) cited articles. For example, in the second/third paragraph, where the concepts of DDA/DIA are introduced or clinical studies that make use of proteomics are discussed. In addition, the advantages that DIA offers over DDA for PTM analyses should be briefly mentioned and the respective work be cited.
- Line 114: 'Three trends can be observed in current DIA analysis strategies: a) using spectral libraries generated by analysing pre-fractionated DDA runs': the authors should mention that fractionated samples can also be acquired by DIA (and processed using an in silico generated spectral library)

Results:

Although this manuscript targets the proteomics community as readership, an appropriate introduction of the employed DIA workflows would be needed to better follow the setup (e.g. what is an experiment-specific spectral library, what is a GPF-refined library? what is a predicted library?) Alternatively, this could be included in the introduction.

In general, the experimental setup and the rationale for the setup should be explained in more detail.

It is not clear what the groups for statistical testing are and what is being tested (E.coli/human proteins?) The authors should also better explain the experimental setup (why four spike-in groups?)

How does GPF work? What does it mean to refine a library?) The authors should also comment on the acquisition order of samples.

The authors directly compare protein outputs from the different software with each other. However, this might be misleading as the protein grouping algorithms in the employed software are substantially different. The better way for comparing identification numbers would be to count quantified peptide precursors (or peptides). E.coli IDs and Human IDs in should be split in all figures.

For assessing false positives and true negatives, the authors test whether human proteins are differentially (and significantly) regulated between the two spike-in conditions. However, as the groups consist of 23 lymph nodes each (derived from 23 individual patients), and proteins can be present in different quantities, this setup is not suitable for such a comparison.

Methods:

More details should be provided in all section and especially on the FFPE tissue sample preparation. Only part of the lysis protocol is described but what about protein precipitation, protein concentration measurements, and peptide clean-up? Also the high pH reversed-phase peptide fractionation should be explained in more detail.

Reviewer #1 (Remarks to the Author):

The authors Fröhlich et al present an interesting approach to benchmarking using bootstrapping drawn from a large dataset. This enables the authors to better estimate parameters for which tool combinations work best in their background, and I think this is clever. In general, the work is well written, considered, and executed.

I found the abstract somewhat misleading and detracting from the work. The abstract currently presents the central advances of this work as benchmarking with a large dataset, the evaluation of 2M workflows, and some very narrow suggestions. To me, this manuscript is much more interesting than that! I recommend that the authors rewrite the abstract in the following ways:

a) It's easy to get a large dataset for testing and evaluation. The main innovation of this work is to develop a test dataset structured to generate bootstrapped smaller datasets, allowing for more robust statistical evaluation of different combinations of libraries, tools, etc. Rather than focus on the size of the dataset/number of evaluations, the abstract should discuss both the bootstrapping approach and how it helps enable more reliable assessments.

b) The take-home messages in the abstract seem far too specific, given the breadth of the evaluations. The abstract says that the specific combination of GPF/DIA-NN + DIA-NN + SAM produces reliable measurements, but realistically, all of the tested methods do that to one degree or another. Additionally, this statement ignores that GPF/Prosit + Skyline produced more quantified proteins than any of the other methods. To me, clearer take-home messages are (1) that GPF-based libraries improved the analysis of all of the software tools (and not just the ones they were designed for, e.g. EncyclopeDIA and Skyline), and (2) that non-parametric statistical testing consistently performed the best at detecting truly changing proteins.

Answer: We thank the reviewer for the overall positive assessment. We have rewritten the abstract taking the recommendations provided by the reviewer into account. In particular, we now emphasize our test dataset from which multiple bootstrap datasets can be generated to enable more robust statistical evaluations. We also included a more differentiated assessment of the results evaluating the different workflows. With this, we think the abstract better describes the strengths of this benchmark paper and the novelty and innovative aspect of our approach.

c) I think the additional take-home message about missing values and false-positive quantification is interesting, but also likely due to the parameter settings for how the authors ran the different software tools (see below). I recommend dulling this message to not discuss performance of specific tools, but rather indicate that it is an important consideration or source of variability.

Answer: As explained in more detail below, we have addressed the issue of the parameter settings to run the different software tools by implementing the TRIC tool into the Galaxy environment and tested the parameter `--max_fdr_quality` with 5 % FDR and 1 % FDR, and by producing a MSstats protein level summarized output for the Skyline output. Accordingly, we have revised the message and included the different handling of missing values as an additional source of variability.

I recommend the authors consider the following additional points:

1) Page 8 line 280: The authors discuss how DIA-NN and Spectronaut report 0s for missing values, while Skyline and OpenSwath report background integration. These are just default choices for each software, and those choices should be made on a dataset-by-dataset (or question-by-question) basis. For example, you can adjust OpenSwath's reporting for background integration by adjusting the `--max_fdr_quality` setting in TRIC. In Skyline, researchers typically filter out background signals based on other features, such as precursor isotope dot product or spectral correlation to the library. Missing values are an important point to discuss, but I suspect that all of these software tools have "toggles" to turn reporting of background integration on or off based on the researcher's use cases.

Answer: We thank the reviewer for this very helpful and insightful comment. To better understand the impact of TRIC parameter adjustment, we have implemented the TRIC tool into the Galaxy environment, so that it can be used by the general public. We have tested the recommended parameter: `--max_fdr_quality` with 5 % FDR and 1 % FDR and have included the results as supplementary Figure S6. We additionally discuss the impact of this parameter and its implications for a user of OpenSwath with regard to the handling of missing values in the Results (line 293-302) and Conclusion (line 596) section. Based on these results, we have included the revised message that missing values are handled differently by different software tools as requested.

We would like to underline that Skyline is not able to perform protein summarization (returns NAs) when different numbers of precursors are present for a protein over multiple samples, which is the case for most proteins after q Value filtering on precursor level.

We have contacted the developers of Skyline and have provided them with one of our analysis files. We agreed with the developers that protein summarization should be performed independently of Skyline. As Skyline partially implements the MSstats tool to achieve this, we exported the precursor level output (with the q-Values derived by the mProphet model) from Skyline and performed protein summarization using MSstats.

As this is the recommendation of the Skyline developers, we chose to replace the original output of Skyline with the MSstats protein level summarized output.

2) Page 10 line 330: The authors discuss using bootstrapping with replacement. I understand the reasoning, but the authors should discuss why they chose to do this and how this could be dangerous, since drawing duplicate samples is possible and may affect the success of downstream tools (e.g. this might help explain in part why non-parametric stats perform so much better than parametric stats, which assume an appropriate level of randomness in the data).

Answer: As suggested, we have adapted the manuscript to better present the rationale and background for the bootstrapping approach. We have added the following statement to the manuscript

“Although bootstrapping is a well established technique, it is known that it can introduce a bias for small sample sizes (i.e. in non-asymptotic settings). For a fair comparison of performances it is most important that this potential bias is shared among the different statistical tests. This is, indeed, the case for the exemplarily chosen samples sizes 3, 6, 13, 23 (Supplementary Figure S19).”

In this figure we show with the example of "DIA-NN (GPF DIA-NN)" how for different sample sizes the proportion of unique samples relates to the pAUC for the different statistical tests. Indeed, for all sample sizes the pAUC increases together with this proportion. This behavior, however, is not unique for parametric tests but is the case for all statistical tests.

Furthermore, the inferior performance of the simple non-parametric Wilcoxon-Mann-Whitney test, which we have added in the revision, suggests that the good performance of SAM and ROTS is rather due to them inherently performing permutations than to their non-parametric nature in general.

3) Page 11 line 356-361: The evaluation of "Combined" vs "Intersection" protein lists is confusing, especially since the "DIA-NN Predicted" workflow seems to detect a very different set of proteins than the rest of the tools (Supp Fig 5). The "Intersection" list seems sensible, but it's hard to reconcile the 11.5K proteins detected in "Combined" given that no single tool consistently detected more than 7K proteins. These lists should be clarified with respect to Fig 2.

Answer: We thank the reviewer for pointing this out. For a better overview over the protein reference lists used, the supplementary figure S9 has been added and the results section has been streamlined to highlight the importance of using different reference protein lists (even if not all entries are validated by multiple workflows in the case of the Combined protein reference list).

4) Conclusions section: I think some discussion is needed of how this dataset of MS/MS runs is similar to more common Human experiments, but also illustrate the ways that it differs. In particular, the authors should indicate that these settings/results may not apply to other types of samples, such as biofluids, cell lines, etc due to the differences in complexity. Additionally, Human+Ecoli may not necessarily be representative of what Human samples are alone.

Answer: We have added the following section at the beginning of the results when introducing the experimental design:

"The experimental design of our study is aimed at investigating highly complex human samples, such as tissue lysates. Other proteomics studies such as plasma proteomics may display a lower complexity (reference Tanaka et al. 2020).

Using such a spike-in design provides ground truth knowledge of certain data properties. However, it also represents an additional complexity usually not present in a human sample. We have carefully considered this and think that using real patient derived samples with an intermediately complex spike-in such as *E. coli* provides a good trade-off between knowing the ground truth of the dataset while still preserving relevant data properties to access biomarker discovery strategies."

5) Methods section: The tools section regularly makes use of "Default" or "Recommended" settings. These settings should be clarified in the manuscript in case those recommendations change in the future.

Answer: We thank the reviewer for this comment and have added the setting parameters, which are necessary to repeat our analyses. We also provide the version number of each software used. For most software options, recommended settings only change with updates.

Reviewer #2 (Remarks to the Author):

The authors present a novel report on a large benchmarking study comparing different Data Independent Acquisition (DIA) proteomics data analysis strategies using a cohort of patient samples. Authors used different amounts of Ecoli protein in patient samples to create populations of known concentration samples which will aid in assessment of effectiveness of different data analysis strategies. DIA data analysis is heavily dependent on spectral library generation both empirically and in-silico, hence authors examined four broadly used data analysis software for both spectral library generation and DIA analysis to evaluate performance of each one. Using bootstrapping technique authors were able to evaluate how the sample size, normalization and all the subsequent statistical test are affecting the ability of the data analysis workflow to determine the correct ratio of Ecoli to human proteins.

DIA proteomics is quickly becoming a mainstream choice of analysis for increased identification and quantitation of proteins in a complex sample. This study has highlighted the need for benchmarking different DIA data analysis strategies especially using a patient cohort with inherent heterogeneity.

The authors have chosen appropriate methodologies to answer the main question of this study by choosing proper spiked in clinical samples for this study, performing all the sample preparation, LC/MS analysis and all the subsequent steps for data analysis and comparison of different spectral library generation workflows and analysis software packages. Authors have provided details of their experimental workflow and data analysis. Results are presented clearly and demonstrate the findings of authors in a logical manner.

Minor comments:

1) In lines 367-369 the following sentence is missing a word "In our study, the reference protein list based on which the ROC is generated represents an additional ?? that impacts the outcome of our comparisons."

Answer: We thank the reviewer for pointing this out.

The sentence has since been replaced to generally streamline the section of the manuscript.

2) Three out of the four data analysis software packages are using iRT peptides for retention time normalization. It would be beneficial if authors could comment about the effect of using iRT peptides for retention time normalization in the software packages which use it versus software package like DIA-NN which does not use iRT peptides. Do authors think that iRT peptides in any way, could affect identifications of peptides in a negative manner?

Answer: We thank the reviewer for this comment.

As iRT peptides are a well established retention time standard that was specifically developed not to interfere with the LC-MS/MS measurements, we chose to include iRT peptides into our experimental setting to also enable their usage during data analysis. The overall good performance of DIA-NN, which to our knowledge did not use the iRT peptides for alignment indicates that iRT peptides are not strictly necessary for a good performance and their effect on the quality of measurement seems to be negligible. To introduce the usage of iRT peptides, we have now included a brief section in the revised manuscript (line 210).

Off note, we often use the iRT peptides for troubleshooting LC-related issues and even if the quality of measurements is not positively influenced by iRT peptides, we think that they overall increase reproducibility for our LC-MS/MS measurements.

Reviewer #3 (Remarks to the Author):

In this study by Frohlich et al, the authors use protein digests derived from lymph nodes mixed with E.coli peptides to test a number of DIA software in combination with different spectral libraries and downstream data processing methods. The aim of the work is to determine the best DIA mass spectrometry and data analysis workflow for clinical samples. The authors claim to determine a generalizable workflow for analyzing clinical samples by DIA mass spectrometry. However, as the authors claimed themselves, clinical samples are very heterogeneous in terms of protein expression levels and they can also be contaminated by e.g. cells from neighboring tissues in case of FFPE or by red blood cells in case of plasma). It is therefore difficult to come up with an all-in-one solution for clinical sample analysis by DIA mass spectrometry. While a general guideline for DIA mass spectrometry could potentially be helpful, the experimental setup of the present study is very poorly explained and it is difficult to grasp what was actually done.

The work, as currently presented, is not suitable to target a broad readership as it is full of technical terms that are not properly explained (e.g. refinement of a spectral library, sparsity reduction).

Overall, the work might be relevant for a more specialized Journal but it is of limited novelty concerning both data processing and data analysis (all standard ways). The study is limited to testing a number of library/software/normalization/statistical tests combinations but lacks critical assessment of the results and data interpretation. Besides similar benchmark studies have already been published (e.g. <https://doi.org/10.1021/acs.jproteome.1c00490>).

Answer: We thank the reviewer for pointing out similarities to other DIA benchmarking studies. We wish to point out that the aforementioned study does not address inter-individual proteome heterogeneity, covers a smaller scale of samples, and does not use multiple protein references, which are necessary to fully grasp the advantages and disadvantages of different DIA analysis strategies.

A hallmark of DIA is the ability to perform proteome profiling of patient cohorts; this ability motivated us to specifically set up a benchmarking study with a dataset that includes the protein number and composition of a human tissue sample and covers inter individual proteome heterogeneity, while also providing a comprehensive evaluation of different DIA analysis workflows.

Abstract:

It is not clear to me why the 92 lymph node samples result in 118 LC-MS/MS runs? I assume the library files were factored in?

Answer: We agree that the wording in the abstract needed to be improved to better explain the experimental setup

We have therefore rephrased the abstract in line with the comments of reviewer #1 focusing more on the bootstrap approach and on generally observable patterns in the results, and less on the individual results of single software solutions. Additionally, we extended the explanation and rationale of the experimental setup at the beginning of the Results section.

Introduction:

- The intro could be more concise and would benefit from some (more) cited articles.

Answer: We have included more cited articles in the Introduction to strengthen our key points of the introduction we wanted to make.

- For example, in the second/third paragraph, where the concepts of DDA/DIA are introduced or clinical studies that make use of proteomics are discussed. In addition, the advantages that DIA offers over DDA for PTM analyses should be briefly mentioned and the respective work be cited.

Answer: We fully agree that the advantage of DIA over DDA can be discussed in greater detail! We have therefore included studies that show the superiority of DIA when analyzing phospho PTMs as an example (line 90). However, as PTM analyses have not been the focus of this manuscript, we would rather not further expand on PTM analyses employing DIA.

- Line 114: 'Three trends can be observed in current DIA analysis strategies: a) using spectral libraries generated by analyzing pre-fractionated DDA runs': the authors should mention that fractionated samples can also be acquired by DIA (and processed using an in silico generated spectral library)

Answer: We fully agree with the reviewer and now mention the possibility to use DIA to measure pre-fractionated samples (line 127). In our workflow we chose DDA for the pre-fractionation measurements, as this allowed us to include MaxQuant and MSFragger into our library generation pipeline. MaxQuant is considered the gold standard in many proteomics analyses and we therefore wanted to include this important software tool. Secondly, we have measured a gas phase fractionation mastermix in DIA mode to refine a predicted library.

Results:

Although this manuscript targets the proteomics community as readership, an appropriate introduction of the employed DIA workflows would be needed to better follow the setup (e.g. what is an experiment-specific spectral library, what is a GPF-refined library? what is a predicted library?) Alternatively, this could be included in the introduction.

Answer: We thank the reviewer for this insight and have included in the Introduction a more detailed explanation of what a spectral library is and how it can also limit the results from a DIA data analysis (line 80).

In general, the experimental setup and the rationale for the setup should be explained in more detail.

It is not clear what the groups for statistical testing are and what is being tested (E.coli/human proteins?) The authors should also better explain the experimental setup (why four spike-in groups? How does GPF work? What does it mean to refine a library?) The authors should also comment on the acquisition order of samples.

Answer: We have included a more detailed overview of the experimental design at the beginning of the Results section. We have also included an explanation of the measurement order of samples (line 218), for which we used block randomization in conjunction with an increasing order of spike-in concentration of *E. coli* peptides to prevent carry over of *E. coli* peptides into the human only condition.

The authors directly compare protein outputs from the different software with each other. However, this might be misleading as the protein grouping algorithms in the employed software are substantially different. The better way for comparing identification numbers would be to count quantified peptide precursors (or peptides).

Answer: We fully agree that identification numbers on precursor levels are of great interest to the readership and we have therefore included ID numbers, quantification distributions and variance distributions on precursor level in the additional supplementary figures S3 and S7a. Furthermore, we have improved the comparability of the protein grouping by doing protein summarization of the Skyline output using MSstats, which allows for a more precise FDR control filtering on precursor and protein level.

E. coli IDs and Human IDs in should be split in all figures.

Answer: We have split the figures wherever possible:

Figure 2

Supplementary Figures S4 & S5

Supplementary Figure S8

Supplementary Figures S10 & S11

In our opinion, Figure 3A & S6 convey a clearer message using the color coding instead of separating human from *E. coli* proteins into different figures.

For assessing false positives and true negatives, the authors test whether human proteins are differentially (and significantly) regulated between the two spike-in conditions. However, as the groups consist of 23 lymph nodes each (derived from 23 individual patients), and proteins can be present in different quantities, this setup is not suitable for such a comparison.

Answer: We respectfully disagree with this assessment. Using bootstrapping in combination with the high number of replicates, the numbers are high enough to allow for the quantification of the false discovery of differentially abundant proteins.

For example, for the investigation of false positives:

While a single comparison of e.g 3 random human samples with 3 other random human samples might yield (real) differentially abundant human proteins, 100 bootstrap analyses of different + random 3 vs 3 human samples cannot consistently yield differentially abundant proteins.

Methods:

More details should be provided in all sections and especially on the FFPE tissue sample preparation. Only part of the lysis protocol is described but what about protein precipitation, protein concentration measurements, and peptide clean-up? Also the high pH reversed-phase peptide fractionation should be explained in more detail.

Answer: We thank the reviewer for pointing out missing information in the Methods section. We have added more information where we think this might be helpful for others who want to use this benchmark dataset or repeat the experiment. We have tried to mimic the depth of details by other studies e.g. doi.org/10.1038/s41467-020-15346-1.

The S-Trap protocol effectively replaces protein precipitation and peptide clean-up. We have briefly summarized the steps in the Methods section (line 667) and provide a reference for a very detailed study that focuses on the validation of the S-Trap protocol.

REVIEWERS' COMMENTS

Reviewer #1 (Remarks to the Author):

The authors Fröhlich et al present an updated article describing their benchmarking approach using bootstrapping. I feel the authors have satisfactorily addressed all of the reviewers concerns. Additionally, the abstract and conclusions now indicate much more generalizable ways to improve DIA analysis in other studies, further improving the impact of the work.

Reviewer #3 (Remarks to the Author):

The authors answered most of my concerns. However, a few things need better explanation or corrections/additions. In general, I would recommend making the manuscript more concise. This should be done especially for the introduction, but also for the 'data analysis results' section in the results part.

Here a list of my recommended corrections:

Abstract:

Lines 45-49: unclear, please rephrase. Moreover, describe what the reference lists are. Alternatively introduce later with a better explanation (I would prefer the latter option)

Introduction:

The section where spectral libraries are introduced could be more concise and it would be beneficial to introduce the concept of library refinement.

Line 59: typo ('allow' instead of 'allow')

Lines 62-63: rephrase (typos)

Lines 85-87: rephrase

Results:

Lines 293-13: make this more concise. One or two sentences are sufficient to present this

Lines 193-200: this section should be more concise as it interrupts the flow of thoughts'

Lines 214-218: remove (not relevant for the this section) and describe in the methods instead

Line 234: Note that using DIANN with at predicted library is the same as SN's directDIA option

Lines 230-42: although the authors added precursor numbers in the supplemental figures, it should be stated that the protein grouping algorithms of the employed software differ, meaning that the number of quantified proteins with the different software cannot directly be compared with each other.

Line 273: typo ('have' instead of 'has')

Lines 301-02: the TRIC tool should be explained

Lines 321-22: rephrase

Line 420: typo ('are' instead of 'is')

Lines 443-49: It should be noted that the analysis in DIANN was done without the 'match-between-function', which is the standard setting in newer version releases. The authors could add one sentence stating that the obtained results should be revisited when using newer software versions

Line 453: 'as follows' instead of 'in the following'

Answers to Reviewer 3:

We thank the reviewer for his overall positive feedback and have considered his suggestions in the following:

Abstract:

Lines 45-49: unclear, please rephrase. Moreover, describe what the reference lists are. Alternatively introduce later with a better explanation (I would prefer the latter option)

Answer:

We agree with the reviewer that more details may help to make the abstract clearer. However, due to the restriction of word count for the abstract (150) it is not possible to go more into detail. We also think that the role of references lists is extremely crucial and while we do not have the opportunity to explain in detail what they are, we want to imply that the picture of “best performing” is more complex upon closer investigation.

Introduction:

The section where spectral libraries are introduced could be more concise and it would be beneficial to introduce the concept of library refinement.

Line 59: typo ('allow' instead of 'allow')

Answer:

This has now been corrected.

Lines 62-63: rephrase (typos)

Answer:

This has been corrected

Lines 85-87: rephrase

"As retention time, relative fragment intensity and the proteins which are present may differ between instruments and experiments, often experiment specific spectral libraries are generated for an individual project."

Answer:

We have rephrased the paragraph to be more concise in the introduction of spectral libraries.

Results:

Lines 293-13: make this more concise. One or two sentences are sufficient to present this

Answer:

We are not sure, which lines the reviewer is referring to.

We assume that the reviewer refers to lines 193-213.

If this is the case:

Rev#1 has asked us to include such a detailed discussion in the manuscript and we think that the placement of this discussion in the manuscript is justified.

If the reviewer does indeed refer to the content starting at line 293:

We do not think that we can shorten this paragraph without losing vital information.

Lines 193-200: this section should be more concise as it interrupts the flow of thoughts'

Answer:

We have shortened this section to streamline the message of the paragraph.

Lines 214-218: remove (not relevant for the this section) and describe in the methods instead

Answer:

We understand the concern of the reviewer that this section is somewhat lengthy. However, it is essential to describe the proteomics dataset as a foundation to the subsequent evaluation of analysis workflows. Respectfully, we wish to refrain from shortening this paragraph as we consider such shortening to reduce comprehensibility of the entire study. We sincerely hope that the reviewer understands our reasoning.

Line 234: Note that using DIANN with at predicted library is the same as SN's directDIA option

Answer:

We now state that also a predicted library combination with DIA-NN does not require additional experimental evidence for library generation

Lines 230-42: although the authors added precursor numbers in the supplemental figures, it should be stated that the protein grouping algorithms of the employed software differ, meaning that the number of quantified proteins with the different software cannot directly be compared with each other.

Answer:

We thank the reviewer for this suggestion and added this to the manuscript.

Line 273: typo ('have' instead of 'has')

Answer:

This has been corrected.

Lines 301-02: the TRIC tool should be explained

Answer:

We now briefly describe the functionality of the TRIC tool. We would not go into more detail as this would lead to a less concise paragraph and would potentially interrupt the flow of argument. We have included the reference of the TRIC publication for interested / versed readers.

Lines 321-22: rephrase

Answer:

This has now been rephrased.

Line 420: typo ('are' instead of 'is')

Answer:

We think that the subject of the sentence is list (singular) and therefore should not be changed.

Lines 443-49: It should be noted that the analysis in DIANN was done without the 'match-between- function', which is the standard setting in newer version releases. The authors could add one sentence stating that the obtained results should be revisited when using newer software versions

Answer:

We agree with the reviewer and have added this to the paragraph.

Line 453: 'as follows' instead of 'in the following'

Answer:

This has now been changed accordingly.